# TEQUILA-seq: a versatile and low-cost method for targeted long-read RNA sequencing

Feng Wang[1,6], Yang Xu[1,2,6], Robert Wang[1,2,6], Beatrice Zhang [1], Noah Smith[1], Amber Notaro[1], Samantha Gaerlan[1], Eric Kutschera [1], Kathryn E. Kadash-Edmondson[1], Yi Xing [1,3,4] ✉ & Lan Lin [3,5] ✉

Long-read RNA sequencing (RNA-seq) is a powerful technology for transcriptome analysis, but the relatively low throughput of current long-read sequencing platforms limits transcript coverage. One strategy for overcoming this bottleneck is targeted long-read RNA-seq for preselected gene panels. We present TEQUILA-seq, a versatile, easy-to-implement, and low-cost method for targeted long-read RNA-seq utilizing isothermally linear-amplified capture probes. When performed on the Oxford nanopore platform with multiple gene panels of varying sizes, TEQUILA-seq consistently and substantially enriches transcript coverage while preserving transcript quantification. We profile full-length transcript isoforms of 468 actionable cancer genes across 40 representative breast cancer cell lines. We identify transcript isoforms enriched in specific subtypes and discover novel transcript isoforms in extensively studied cancer genes such as *TP53*. Among cancer genes, tumor suppressor genes (TSGs) are significantly enriched for aberrant transcript isoforms targeted for degradation via mRNA nonsense-mediated decay, revealing a common RNA-associated mechanism for TSG inactivation. TEQUILA-seq reduces the per-reaction cost of targeted capture by 2-3 orders of magnitude, as compared to a standard commercial solution. TEQUILA-seq can be broadly used for targeted sequencing of full-length transcripts in diverse biomedical research settings.

Alternative isoform variation is prevalent in eukaryotic transcriptomes and substantially diversifies eukaryotic gene products[1,2]. Various RNA-level regulatory processes, such as alternative splicing (AS) and alternative polyadenylation, can produce multiple transcript isoforms from a single gene[3,4]. Resulting transcript isoforms may differ in RNA localization, stability, and translational efficiency, or may code for protein isoforms with distinct structures and functions[5,6].

Over the last decade, short-read RNA sequencing (RNA-seq) has been broadly used as the standard approach for transcriptome analysis[7]. Due to its read length, however, short-read RNA-seq is limited in its ability to resolve full-length transcript isoforms and complex RNA processing events[8]. By contrast, long-read sequencing platforms, such as Pacific Biosciences (PacBio) and Oxford Nanopore Technologies (ONT), can generate reads longer than 10 kb and directly sequence full-length transcript molecules end-to-end[9,10]. However, a major

[1]Center for Computational and Genomic Medicine, Children's Hospital of Philadelphia, Philadelphia, PA, USA. [2]Graduate Group in Genomics and Computational Biology, University of Pennsylvania, Philadelphia, PA, USA. [3]Department of Pathology and Laboratory Medicine, University of Pennsylvania Perelman School of Medicine, Philadelphia, PA, USA. [4]Department of Biomedical and Health Informatics, Children's Hospital of Philadelphia, Philadelphia, PA, USA. [5]Raymond G. Perelman Center for Cellular and Molecular Therapeutics, Children's Hospital of Philadelphia, Philadelphia, PA, USA. [6]These authors contributed equally: Feng Wang, Yang Xu, Robert Wang. ✉e-mail: xingyi@chop.edu; linlan@chop.edu

limitation of long-read sequencing platforms is that their throughput is multiple orders of magnitude lower than that of short-read platforms (Illumina, in particular)[11]. This limitation poses a major bottleneck for transcriptome analysis, which requires high sequencing coverage to accurately quantify transcripts and measure isoform proportions, as well as sensitively discover low-abundance transcripts.

Targeted sequencing, which involves enriching specific sequences of interest, provides a useful strategy for substantially enhancing transcript coverage of a preselected gene panel. Several approaches have been developed for targeted long-read RNA-seq. Single or multiplex long-range RT-PCR amplification utilizes primer pairs placed at terminal exons to amplify target transcripts, followed by long-read sequencing[12]. However, this approach may fail to enrich transcripts with novel alternative first or last exons, and may not scale up to large gene panels due to issues of primer cross-reactivity and amplification bias. Hybridization capture-based enrichment[13,14] using biotinylated capture oligos such as RNA Capture Long Seq (CLS)[15] is an efficient method for targeted long-read RNA-seq. Nevertheless, commercially synthesized biotinylated capture oligos are expensive and can only be used for a limited number of reactions, making the per-sample cost high for each targeted capture. Sheynkman et al. recently described an alternative hybridization capture-based approach called ORF Capture-Seq that uses directly synthesized biotinylated capture oligos from open reading frame (ORF) clones[16]. However, accessing and operating the human ORFeome library is resource- and time-consuming, and this approach cannot be applied to genes not included in the human ORFeome library.

To address these limitations, we have developed TEQUILA-seq (**T**ranscript **E**nrichment and **Q**uantification **U**tilizing **I**sothermally **L**inear-**A**mplified probes in conjunction with long-read **seq**uencing). A key innovation in TEQUILA-seq is that it uses nicking-endonuclease (nickase)-triggered isothermal strand displacement amplification (SDA) to synthesize large quantities of biotinylated capture oligos from an array-synthesized pool of non-biotinylated oligo templates. This strategy for synthesizing capture oligos makes TEQUILA-seq highly cost-effective and scalable for large gene panels and sample sizes. To benchmark its performance, we performed TEQUILA-seq using the ONT platform for multiple gene panels of varying sizes on synthetic RNAs or human mRNAs. To illustrate its biomedical utility, we applied TEQUILA-seq to profile full-length transcript isoforms of 468 actionable cancer genes across a broad panel of 40 breast cancer cell lines representing distinct intrinsic subtypes. We identified transcript isoforms enriched in specific subtypes and discovered novel transcript isoforms in extensively studied cancer genes such as *TP53*. Among cancer genes, tumor suppressor genes (TSGs) were significantly enriched for aberrant transcript isoforms targeted for degradation via mRNA nonsense-mediated decay, revealing a common RNA-associated mechanism for TSG inactivation. TEQUILA-seq can be broadly used for targeted sequencing of full-length transcripts in diverse biomedical research settings. Moreover, TEQUILA probes are compatible for both targeted RNA and DNA sequencing, on both long-read and short-read sequencing platforms. The ability to easily generate large quantities of biotinylated capture oligos for any target panel at a low cost and a high efficiency can facilitate large-scale and population-level studies for a wide range of basic, translational, and clinical applications.

## Results
### Overview of TEQUILA-seq
We developed TEQUILA as a versatile, easy-to-implement, and low-cost approach for generating large quantities of biotinylated capture oligos for any gene panel (Fig. 1a). First, single-stranded DNA (ssDNA) oligos are designed to tile across all annotated exons of target genes and synthesized using an array-based DNA synthesis technology. Next, TEQUILA probes are amplified from ssDNA oligo templates in a single

pool using nickase-triggered SDA with universal primers and biotin-dUTPs. SDA enables isothermal amplification of internally biotinylated oligos through repeated cycles of nicking and extension reactions using a strand displacement DNA polymerase and pre-designed nickase-targeted nicking sites. This process allows large quantities of capture oligos to be generated from starting templates. The resulting pool of TEQUILA probes can be used to capture full-length cDNA molecules of genes of interest. Because of the use of a low-cost ssDNA oligo pool and the large probe synthesis output, TEQUILA substantially reduces the setup and per-reaction costs of targeted capture compared to commercial methods (Supplementary Data 1, 2). For example, a custom set of biotinylated capture oligos from Integrated DNA Technologies (IDT) for a 6000-probe panel is $13,000 for 16 reactions (~$813/reaction). By contrast, the setup cost of TEQUILA probe synthesis for the same 6000-probe panel is $3,086 ($1,820 for oligo pool), and this pool can potentially be used to synthesize TEQUILA probes for 6,250 to 25,000 reactions, at $0.31–$0.53/reaction when considering the costs of reagents and consumables.

When coupled with long-read RNA-seq, TEQUILA-seq is designed to provide high coverage of full-length transcripts to facilitate comprehensive discovery and accurate quantification of transcript isoforms (Fig. 1b). Briefly, full-length cDNAs are synthesized from poly(A)+ RNAs by reverse transcription and PCR amplification. TEQUILA probes are then hybridized to cDNAs. Upon capture and washing, cDNA-to-probe hybrids are immobilized to streptavidin magnetic beads, whereas unbound cDNAs are washed away. Captured cDNAs are further amplified by PCR and subjected to nanopore 1D library preparation and sequencing. Finally, TEQUILA-seq data are analyzed by our ESPRESSO software, designed for robust transcript analysis using long-read RNA-seq data[17].

### TEQUILA-seq enriches target transcripts comparably to a standard commercial solution
We assessed the capture efficiency and target enrichment of TEQUILA-seq relative to xGen Lockdown (IDT) probe-based target enrichment and sequencing (hereafter referred to as xGen Lockdown-seq), a standard commercial solution for targeted RNA-seq. We initially designed a test panel of 10 human genes (*DAB1, DLG4, GRIN1, HTT, LRP8, MAPT, NRXN1, NUMB, RBFOX1*, and *SCN8A;* see Supplementary Data 3). These genes were selected because they are known to express long transcripts with complex AS patterns in the brain[18–20]. For this panel, we synthesized TEQUILA probes and ordered xGen Lockdown probes with the same probe sequences at 1x tiling density. We applied both probe sets to the same human brain cDNA sample and generated nanopore 1D sequencing data with comparable sequencing depths (Supplementary Data 4). Estimated abundances of transcript isoforms were nearly identical across all TEQUILA-seq and xGen Lockdown-seq libraries (Supplementary Fig. 1). Compared to whole-transcriptome nanopore RNA-seq data generated on the same brain cDNA sample (i.e., a non-capture control), both TEQUILA and xGen Lockdown probes showed comparable performances in enriching transcripts from the 10-gene panel. Specifically, both methods achieved an on-target rate of ~85% with similar fold enrichment (~280x) (Fig. 1c). Moreover, both methods yielded nearly identical fold enrichment for each target gene (Fig. 1c and Supplementary Fig. 2). Collectively, these results demonstrate that TEQUILA-seq achieves comparable performance in capture efficiency to a widely used commercial solution. Additionally, the estimated abundances of the 10 target genes by both TEQUILA-seq and xGen Lockdown-seq were highly correlated with the estimated abundances by whole-transcriptome 1D cDNA sequencing (Pearson's correlation of 0.985 and Spearman's correlation of 0.976 between TEQUILA-seq and whole-transcriptome 1D cDNA sequencing; Pearson's correlation of 0.987 and Spearman's correlation of 0.988 between xGen Lockdown-seq and whole-transcriptome 1D cDNA sequencing).

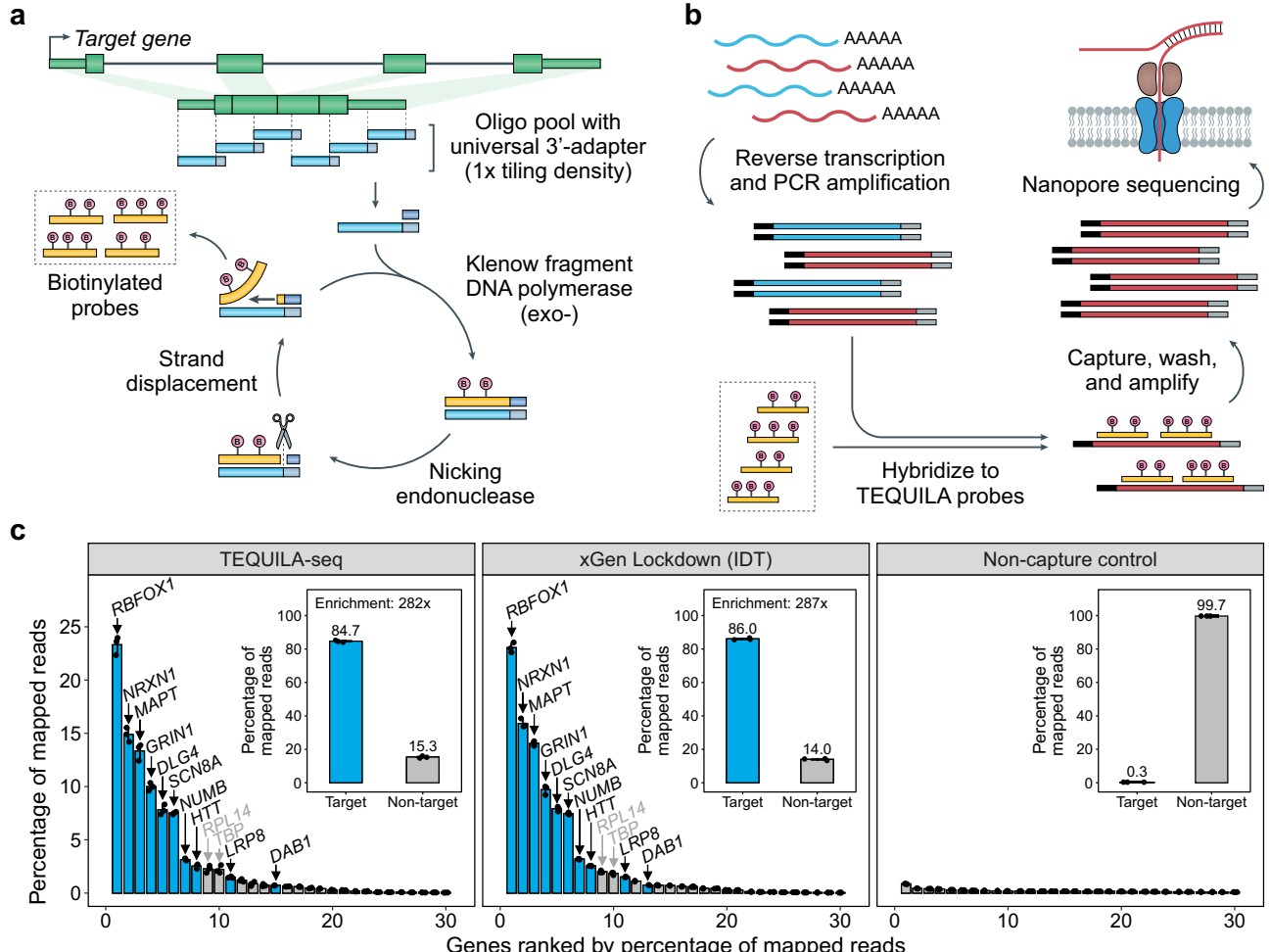

**Fig. 1 | Overview of TEQUILA-seq. a, b** Schematic of TEQUILA-seq. **a** Single-stranded DNA (ssDNA) oligonucleotides are designed to tile across all annotated exons of target genes and synthesized using an array-based DNA synthesis technology. TEQUILA probes are amplified from ssDNA oligo templates in a single pool using nickase-triggered strand displacement amplification with universal primers and biotin-dUTPs. **b** Full-length cDNAs are synthesized from poly(A)+ RNAs by reverse transcription and PCR amplification. TEQUILA probes are then hybridized to cDNAs. Upon capture and washing, cDNA-to-probe hybrids are immobilized to streptavidin magnetic beads, whereas unbound cDNAs are washed away. Captured cDNAs are further amplified by PCR and subjected to nanopore 1D library preparation and sequencing. **c** Comparison of TEQUILA-seq vs xGen Lockdown (IDT) probe-based target enrichment and sequencing. Main graphs: percentage of reads mapped to a given gene (mean ± s.d. of $n = 3$ replicates), for the top 30 genes with the highest number of mapped reads. Insets: percentage of reads mapped to target genes and non-target genes (mean ± s.d. of $n = 3$ replicates). Blue: target genes. Gray: non-target genes. Target gene panel: ten human genes with long transcripts in the brain. All sequencing methods were applied to the same human brain RNA mix from multiple donors (see Methods). Source data are provided as a Source Data file.

## TEQUILA-seq greatly enhances detection and preserves quantification of target transcripts

To evaluate the performance of TEQUILA-seq for transcript detection and quantification, we used RNAs of human SH-SY5Y neuroblastoma cells mixed with synthetic transcripts as spike-ins, including External RNA Controls Consortium (ERCC) standards and Spike-In RNA Variants (SIRVs). TEQUILA-seq was performed using three sets of probes targeting selected ERCC and SIRV synthetic transcripts as well as 221 human genes encoding splicing factors (see below). First, we assessed the extent to which TEQUILA-seq improves the detection of transcript isoforms of target genes by using the ERCC standards. The ERCC standards are 92 synthetic transcripts of unique sequences, and their concentrations span six orders of magnitude[21]. These ERCC transcripts were 281 to 2036 nt in length, largely overlapping with the length distribution of human protein-coding transcripts[22]. We synthesized TEQUILA probes for 46 ERCC transcripts covering the entire ERCC concentration range (Supplementary Data 5). The remaining 46 ERCCs were not targeted and served as controls. Using TEQUILA-seq (48 h of runtime), we were able to detect target ERCC transcripts at concentrations as low as 0.18 amol/µl consistently across three replicates

(≥2 reads per replicate) (Fig. 2a). By contrast, this concentration was 65.1 fold higher at 11.72 amol/µl for whole-transcriptome nanopore 1D cDNA sequencing (48 h of runtime).

To investigate how the detection sensitivity of TEQUILA-seq changes with sequencing depth, we sequenced TEQUILA-seq libraries prepared from the same RNA sample for 4 or 8 h. The 4- and 8-h TEQUILA-seq runs had sequencing depths that were six to eight times shallower than the original 48-h TEQUILA-seq runs (Supplementary Data 6). Nevertheless, target ERCC transcripts were still consistently detected at concentrations as low as 0.18 amol/µl in both the 4- and 8-h TEQUILA-seq runs. Moreover, estimated abundances of target ERCC transcripts in TEQUILA-seq libraries were highly correlated with their initial spike-in concentrations, even with the shallower sequencing depth (Pearson's correlation of 0.97 in 48-h TEQUILA-seq, and 0.95 in 8- and 4-h TEQUILA-seq). By comparison, we obtained lower Pearson's correlation values by whole-transcriptome 1D cDNA sequencing (0.93) and direct RNA sequencing (0.79) (Fig. 2a). These results indicate that the TEQUILA probes enriched all 46 target ERCC transcripts at uniformly elevated levels. Additionally, in the same TEQUILA-seq libraries, the estimated abundances of non-target ERCC transcripts were

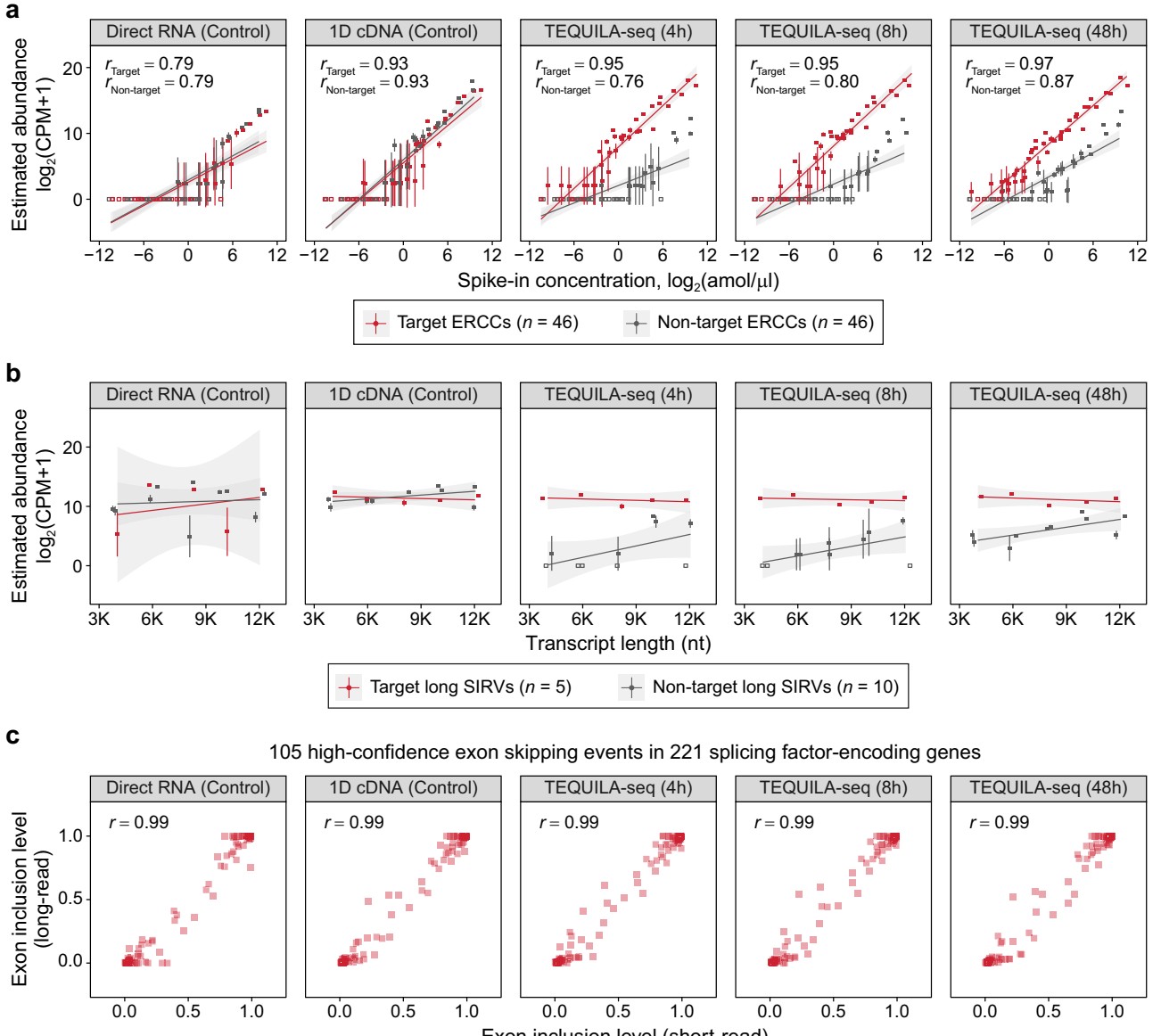

**Fig. 2 | Transcript detection and quantification by TEQUILA-seq. a** Correlation between estimated transcript abundance and spike-in concentration for 92 ERCC transcripts using different sequencing methods. **b** Relationship between estimated transcript abundance and transcript length for 15 long SIRV transcripts of equimolar concentrations using different sequencing methods. In (**a**) and (**b**), each data point is one individual transcript. Estimated abundance is shown as the mean ± s.d. of $n = 3$ replicates. Red: target transcripts. Gray: non-target transcripts. Hollow: undetected transcripts. Pearson's correlation $r$ (**a**) and regression lines (**a**, **b**) were calculated separately for target and non-target transcripts. Gray area: the 95% confidence interval of each regression line. **c** Correlations between estimated exon inclusion levels using different long-read sequencing methods and short-read RNA-seq. Shown are 105 high-confidence exon skipping events identified from 221 splicing factor-encoding genes (see Methods). Each data point represents the mean exon inclusion level ($n = 3$ replicates) for one exon skipping event. In (**a**–**c**), the target gene panel comprises 46 ERCC transcripts, five long SIRV transcripts, and 221 splicing factor-encoding human genes. All sequencing methods were applied to the same RNA sample of SH-SY5Y cells spiked with SIRV-Set 4 spike-in RNA variants that include the ERCC mix and the long SIRVs (see Methods). Source data are provided as a Source Data file.

substantially lower and less correlated (0.76-0.87) with initial spike-in concentrations. Collectively, these results indicate that TEQUILA-seq greatly enhances the detection of target transcripts, even for transcripts with low abundances and in samples with shallow sequencing depth.

Next, we examined whether TEQUILA-seq data exhibit any transcript length-dependent biases. We used a set of SIRVs[23] comprising 15 synthetic transcripts of equimolar concentrations that cover transcript lengths from 4000 to 12,000 nt (hereafter referred to as "long SIRVs"). Our intention behind using long SIRVs was to investigate if there was a preferential enrichment of long transcripts because they had more probes targeting them (e.g., more probes targeting a 12 kb

transcript than a 4 kb transcript). We synthesized TEQUILA probes for five long SIRV transcripts that covered the entire length range of the long SIRV set (Supplementary Data 5). All five targeted long SIRV transcripts had nearly identical estimated abundances across all TEQUILA-seq runtimes (Fig. 2b). These results indicate that the TEQUILA probes enrich target transcripts without exhibiting transcript length-dependent biases.

A potential concern with TEQUILA-seq is that different transcript isoforms of a given target gene may not be enriched at equal levels, thus distorting the relative proportions of transcript isoforms. We reasoned that if TEQUILA probes preserve isoform proportions, then the estimated inclusion levels of alternatively spliced exons within

target genes should remain the same with or without targeted capture. To investigate this issue, we synthesized TEQUILA probes for 221 human genes encoding splicing factors[24] (Supplementary Data 7). These genes are known to undergo extensive AS themselves, as a mechanism to regulate splicing factor activity and function[25–28]. We compared TEQUILA-seq data of this splicing factor gene panel to data generated on the same RNA sample by bulk short-read RNA-seq, as well as by whole-transcriptome nanopore 1D cDNA sequencing and direct RNA sequencing (Supplementary Data 6, 8). Across the 221 splicing factor-encoding genes, the estimated exon inclusion levels for 105 high-confidence exon skipping events (quantified by rMATS[29], see Methods) were highly correlated between short-read RNA-seq and TEQUILA-seq data (Pearson's correlation of 0.99 at 48-, 8-, and 4-h run-times) (Fig. 2c). Similarly, exon inclusion levels estimated using whole-transcriptome nanopore 1D cDNA or direct RNA sequencing were also highly correlated with estimates made by short-read RNA-seq (Pearson's correlation of 0.99). These results indicate that TEQUILA-seq can preserve the relative proportions of transcript isoforms of target genes.

### TEQUILA-seq profiles full-length transcript isoforms of actionable cancer genes across diverse breast cancer cell lines

To illustrate the biomedical utility of TEQUILA-seq, we performed a TEQUILA-seq analysis of actionable cancer genes in a broad panel of breast cancer cell lines. We synthesized TEQUILA probes for 468 genes interrogated by MSK-IMPACT (Memorial Sloan Kettering-Integrated Mutational Profiling of Actionable Cancer Targets), an FDA-approved diagnostic test for DNA-based mutation profiling of actionable cancer targets[30,31] (Fig. 3a and Supplementary Data 9). As alternative isoform variation is prevalent in breast cancer transcriptomes[32,33], we hypothesized that a TEQUILA-seq analysis could discover RNA-associated mechanisms and novel aberrant transcript isoforms in breast cancer. We analyzed 40 breast cancer cell lines from the ATCC Breast Cancer Cell Panel representing four distinct intrinsic subtypes: luminal, HER2 enriched, basal A, and basal B (Fig. 3a and Supplementary Data 10).

We first assessed the degree to which TEQUILA probes could enrich transcripts of genes in this large 468-gene panel. To this end, we performed TEQUILA-seq and whole-transcriptome nanopore 1D cDNA sequencing (as a non-capture control) for four breast cancer cell lines: HCC1806, MDA-MB-157, AU-565, and MCF7 (Fig. 3b and Supplementary Fig. 3). On-target rates of the 468 genes in TEQUILA-seq data ranged from 62.8 to 71.4%, compared to 2.9 to 3.6% in non-capture control data, demonstrating an average ~20-fold enrichment. We then applied TEQUILA-seq to all 40 breast cancer cell lines, with two experimental replicates per cell line, and obtained on-target rates ranging from 62.3 to 73.7% across cell lines (Supplementary Data 11). Nearly all (462/468) of the genes were detected (counts per million or CPM ≥1) in at least one sample. From the entire TEQUILA-seq dataset of the 40 cell lines, we discovered 3122 annotated and 25,519 novel transcript isoforms of the cancer genes. Although many more novel than annotated transcript isoforms were discovered, the majority of reads (79.4% on average across all samples) that mapped to these genes were from annotated transcript isoforms.

Clustering analysis using isoform proportions of the cancer genes revealed two major clusters: cell lines annotated as luminal and HER2 enriched subtypes clustered together, whereas cell lines annotated as basal A and basal B subtypes clustered together (Fig. 3c). Several outlier cell lines were also observed. Specifically, two pairs of cell lines clustered together as outliers, i.e., MDA-MB-453 and MDA-kb2, as well as AU-565 and SK-BR-3, reflecting the similar origins of cell-line derivation[34,35]. The DU4755 cell line, despite its annotation as the basal B subtype, clustered with the luminal and HER2 enriched subtypes, likely reflecting its controversial subtype classification[36,37].

### TEQUILA-seq identifies breast cancer subtype-associated and tumor aberrant transcript isoforms

Next, we used the TEQUILA-seq data to identify subtype-associated transcript isoforms in the 40 breast cancer cell lines (see Methods). For each intrinsic subtype (luminal, HER2 enriched, basal A, basal B), we compared the mean proportion of a transcript isoform between the subtype-associated cell lines and all other cell lines. At FDR ≤0.05, we identified 54 breast cancer subtype-associated transcript isoforms in 50 genes (Supplementary Data 12). As an example, *DNMT3B* encodes a de novo DNA methyltransferase with important roles in regulating DNA methylation in development and cancer[38,39]. Our results reveal that an alternative transcript isoform (ENST00000348286) was highly expressed in the basal B breast cancer cell lines (Fig. 3d, f). Compared to the canonical transcript isoform (ENST00000328111), three exons (exon 10, 21, and 22) were skipped in the alternative transcript isoform (Fig. 3e). Skipping of exons 21 and 22 disrupts the C-terminal catalytic domain; thus, the encoded protein isoform is enzymatically inactive[40]. To summarize, TEQUILA-seq identified a subtype-associated transcript isoform of *DNMT3B*, which may have a global effect on DNA methylation of the basal B subtype of breast cancer. Two additional examples of subtype-associated transcript isoforms were shown for *FGFR2*[41] (Supplementary Fig. 4) and *SESN1* (Supplementary Fig. 5). The transcript isoform switch of *FGFR2* detected by TEQUILA-seq across breast cancer subtypes is a hallmark of epithelial-to-mesenchymal transition associated with the invasiveness and aggressiveness of breast cancer cells[42]. This transcript isoform switch involves two mutually exclusive exons that encode different versions of the ligand binding domain (Supplementary Fig. 4), generating FGFR2 protein isoforms with distinct ligand binding specificities[43]. The transcript isoform switch of *SESN1* involves mutually exclusive uses of alternative first exons (Supplementary Fig. 5) and has been associated with isoform-specific transcript induction by genotoxic stress in a p53-dependent manner[44].

Besides identifying subtype-associated transcript isoforms, we also used TEQUILA-seq data to identify "tumor aberrant" transcript isoforms. We defined tumor aberrant transcript isoforms as transcript isoforms that are present at significantly elevated proportions in at least one but no more than 4 (i.e., ≤10%) breast cancer cell lines. We note that the tumor aberrant transcript isoforms identified based on this definition were essentially "outlier" transcript isoforms with elevated proportions in a small subset of breast cancer cell lines. To eliminate potential technical artifacts and ensure the reliability of our results, we required that the identified tumor aberrant transcript isoforms must be present as outliers in both replicates of a given cell line, and we applied stringent criteria to filter out potential false positives (Methods). In total, we identified 635 tumor aberrant transcript isoforms in 256 genes, with 66.8% being novel transcript isoforms (Fig. 4a and Supplementary Fig. 6). Comparing aberrant to canonical transcript isoforms of the corresponding genes, we found that transcript isoforms resulting from complex or combinatorial AS events represented the majority (69.1%) of aberrant transcript isoforms (Fig. 4b). Given that complex and combinatorial AS events are challenging to analyze and often overlooked by short-read RNA-seq[8], these results highlight the benefit of interrogating the transcript products of actionable cancer genes by long-read RNA-seq.

Using TEQUILA-seq data, we identified numerous novel aberrant transcript isoforms in extensively studied cancer genes, such as *NOTCH1* and *RB1*. A novel aberrant transcript isoform of *NOTCH1* (ESPRESSO:chr9:9147:301) was found as the predominant transcript isoform in the MDA-MB-157 cell line. This transcript isoform lacks the segment spanning exons 2 to 27 with respect to the canonical transcript isoform of *NOTCH1* (Supplementary Fig. 7). In the HCC1937 cell line, we discovered a novel aberrant transcript isoform of *RB1* (ESPRESSO:chr13:2429:105), which lacks exon 22 with respect to the canonical transcript isoform (Supplementary Fig. 8). We confirmed that the novel aberrant transcript isoforms result from focal genomic

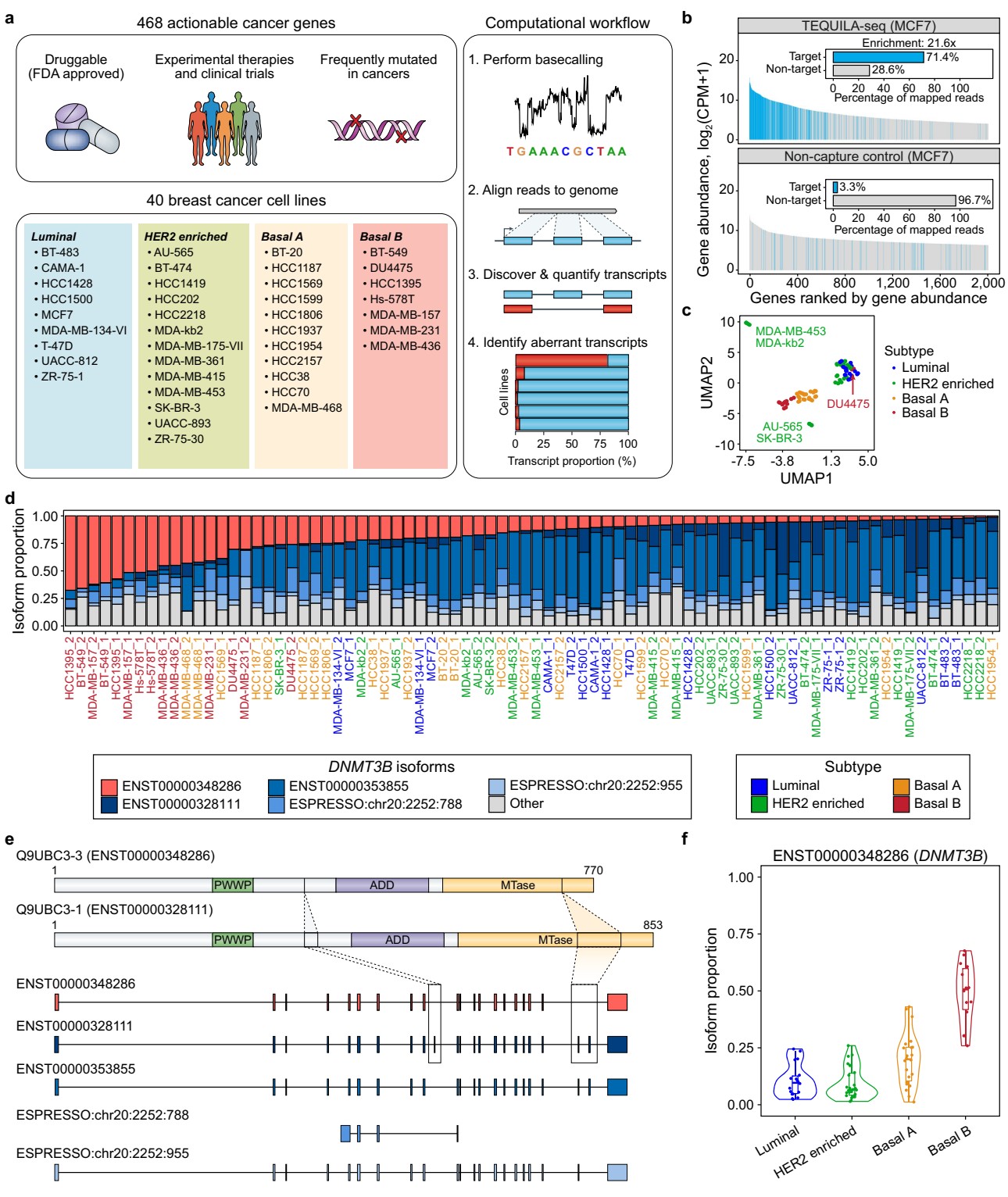

**e** Q9UBC3-3 (ENST00000348286)

**f** ENST00000348286 (*DNMT3B*)

deletions that deleted multiple exons (in *NOTCH1*) or one exon (in *RB1*) from the tumor genome (Supplementary Figs. 7, 8).

**Nonsense-mediated mRNA decay of aberrant transcript isoforms is a common mechanism of tumor suppressor gene inactivation**

Many tumor aberrant transcript isoforms identified by TEQUILA-seq contain premature termination codons (PTCs), which would target the transcript isoform for degradation via nonsense-mediated mRNA decay (NMD)[45]. For example, the tumor suppressor *TP53* encodes a

transcription factor involved in regulating diverse cellular processes, such as cell cycle control, DNA repair, apoptosis, metabolism, and cellular senescence[46,47]. We discovered a novel aberrant transcript isoform of *TP53* (ESPRESSO:chr17:1864:802) as the predominant isoform in the HCC1599 cell line (Fig. 4c). This transcript isoform contains a 568nt retained intron with respect to the canonical transcript isoform of *TP53* (Fig. 4d). The retained intron would introduce an in-frame PTC and target the transcript isoform for degradation via NMD[45]. A second, relatively minor novel *TP53* transcript isoform (ESPRESSO:chr17:1864:391), which uses a novel 3' splice site within the retained

**Fig. 3 | TEQUILA-seq analysis of actionable cancer genes in 40 breast cancer cell lines. a** Summary for TEQUILA-seq analysis of 468 actionable cancer genes in 40 breast cancer cell lines. (Upper left) TEQUILA probes were synthesized for 468 genes interrogated by the MSK-IMPACT gene panel. (Lower left) TEQUILA-seq was performed on 40 cell lines from the ATCC Breast Cancer Cell Panel. These cell lines represent four distinct intrinsic subtypes: luminal, HER2 enriched, basal A, and basal B. (Right) Computational workflow for processing TEQUILA-seq data. **b** Comparison of estimated gene abundances in MCF7 cell line using TEQUILA-seq and whole-transcriptome nanopore 1D cDNA sequencing (non-capture control). Main graphs: top 2000 genes with the highest estimated abundance in each method are shown. Insets: percentage of reads mapped to target genes and non-target genes. Blue: target genes. Gray: non-target genes. **c** UMAP clustering analysis using isoform proportions of all transcript isoforms across 468 genes in 40 cell lines ($n = 2$ replicates). Each data point represents one replicate of a cell line.

**d** Stacked barplots showing proportions of *DNMT3B* transcript isoforms discovered by TEQUILA-seq in 40 cell lines. Red bar: isoform of interest (ENST00000348286); navy bar: canonical isoform (ENST00000328111); lighter blue bars: three other most abundant *DNMT3B* isoforms; gray bars: remaining *DNMT3B* isoforms. **e** Structures of *DNMT3B* protein and transcript isoforms. (Upper) Domain annotations for protein isoforms. PWWP proline-tryptophan-tryptophan-proline domain, ADD ATRX-DNMT3-DNMT3L-type zinc finger domain, MTase methyltransferase domain. (Lower) Transcript structures. Boxes: exons. Line segments: introns. **f** Violin plots (median, interquartile range) showing the distribution of isoform proportions for the *DNMT3B* isoform of interest (ENST00000348286) in different breast cancer intrinsic subtypes. Each data point represents the isoform proportion in a given cell line replicate ($n = 18$ for Luminal, $n = 26$ for HER2 enriched, $n = 22$ for Basal A, and $n = 14$ for Basal B). Source data are provided as a Source Data file.

intron, was also discovered in the HCC1599 cell line (Fig. 4c, d). This transcript isoform is also NMD-targeted. Overall, the discovery of multiple NMD-targeted transcript isoforms is consistent with the low steady-state gene expression level of *TP53* in HCC1599, as measured by TEQUILA-seq (Fig. 4c).

To elucidate the source of these novel *TP53* transcript isoforms, we analyzed the whole-genome sequencing (WGS) data of HCC1599 obtained from the Cancer Cell Line Encyclopedia (CCLE). We found that the HCC1599 cell line harbors an A-to-T mutation that disrupts the 3′ splice site of the retained intron. All WGS reads covering this region contain the same mutation, as the other allele of *TP53* is lost in the tumor genome of HCC1599 through loss of heterozygosity[48]. This splice site mutation and resulting transcript products were further confirmed by RT-PCR and Sanger sequencing (Supplementary Fig. 9). In summary, TEQUILA-seq discovered novel aberrant transcript isoforms of *TP53* in HCC1599, which may contribute to inactivating *TP53* in this cell line.

We should note that the efficiency of NMD varies considerably across transcripts and tissue types depending on various factors such as PTC position[49], the concentration of NMD factors[50], tissue and cell type[51,52], and physiological condition[53]. As a result, PTC-containing transcript isoforms can frequently escape NMD and be readily detected in transcriptome data[49,51]. Interestingly, the cell line with multiple NMD-targeted *TP53* transcript isoforms (HCC1599) was among those with the lowest steady-state *TP53* gene expression levels across the 40 breast cancer cell lines (Fig. 4c), consistent with the expected effect of NMD on steady-state gene expression level. To further confirm that these *TP53* transcript isoforms were NMD-targeted, we treated HCC1599 cells with NMD inhibitor SMG1i (compound 11j). We observed a substantial increase in the isoform abundances of the two PTC-containing transcript isoforms, especially the predominant novel transcript isoform containing the retained intron (Supplementary Fig. 10). These data indicate that these transcript isoforms were indeed targeted and degraded by NMD at the steady state in HCC1599 cells.

The discovery of NMD-targeted aberrant transcript isoforms in *TP53* raises the question of whether this observation represents a recurring RNA-associated mechanism for inactivating tumor suppressor genes in breast cancer. Based on annotations from OncoKB[54] (version September 6, 2022), we categorized the 468 cancer genes analyzed by TEQUILA-seq into three groups: 196 tumor suppressor genes (TSGs), 179 oncogenes (OGs), and 93 "Other" genes. The "Other" category includes genes with context-dependent behavior as either a TSG or an OG, as well as genes with inconclusive roles in the context of cancer. Among genes expressed in at least 10 of the 40 breast cancer cell lines (i.e., average CPM ≥1 between $n = 2$ biological replicates), NMD-targeted aberrant transcript isoforms were significantly more enriched in TSGs (20.9% in TSGs, 9.8% in OGs, and 8.3% in Other; two-sided Fisher's exact test, Fig. 4e). Additionally, the percentages of genes with NMD-targeted aberrant transcript isoforms among genes expressed in each of the 40 breast cancer cell lines were significantly

higher for TSGs than for OGs and Other genes (two-sided paired Wilcoxon test; Fig. 4f). These results suggest that aberrant alternative isoform variation coupled with NMD represents a common mechanism for inactivating TSGs in individual tumors.

## Technical refinements of TEQUILA-seq

Having demonstrated the performance and utility of TEQUILA-seq using multiple target panels, we sought to further assess and refine TEQUILA-seq in multiple aspects. First, we assessed the probe synthesis yield by TEQUILA. We performed a series of probe synthesis reactions for the 468 actionable cancer genes, using varying amounts of template oligo pool ranging from 2 to 10 ng, and varying incubation times ranging from 1 to 16 h. For each of these combinations, TEQUILA probes were synthesized using fixed concentrations of enzymes, dNTPs, and other reaction components as described (see Methods). We quantified the resulting probe yield using Nanodrop (Supplementary Fig. 11). With increasing amounts of template oligo pool, we observed a corresponding increase in probe yield. Moreover, the probe yield increased with longer incubation times, although the rate of increase slowed down after 4 h. This may be explained by the fact that the enzymes and dNTPs used in the probe synthesis reaction became exhausted over time. Overall, the results demonstrate a high probe synthesis yield.

Second, blocking oligos are commonly used in hybridization capture-based enrichment[16,55]. To investigate whether TEQUILA-seq could benefit from the use of blocking oligos, we designed three types of blocking oligos targeting the adapters used in probe synthesis, the primers for cDNA amplification, and oligo(dT)18 with a three-carbon spacer at the 3′ end targeting the poly(A) tail (Methods). We performed TEQUILA-seq of 468 actionable cancer genes on four breast cancer cell lines (HCC1806, MDA-MB-157, AU-565, and MCF7) with or without using blocking oligos. By comparing the results of TEQUILA-seq with or without using blocking oligos, we found that the use of blocking oligos further improved the on-target rates by 2.8 to 5.6% (Supplementary Fig. 12). Overall, our results demonstrate that using blocking oligos can modestly improve the capture performance of TEQUILA-seq.

Finally, we sought to implement the multiplexing capability of TEQUILA-seq. For this purpose, we utilized the ONT native barcoding kit (NBD114) for the R10.4 version of ONT sequencing chemistry. We added sample barcodes and performed multiplexed TEQUILA-seq on four breast cancer cell lines (HCC1806, MDA-MB-157, AU-565, and MCF7) on an R10.4 MinION flow cell. We generated 7.2 million reads. 88.4% of reads can be uniquely assigned to one of the four cell lines with correct sample barcodes, indicating excellent performance of demultiplexing. Compared to TEQUILA-seq of the four breast cancer cell lines without barcoding and multiplexing, TEQUILA-seq with barcoding and multiplexing resulted in similar on-target rates (Supplementary Fig. 13a). Moreover, TEQUILA-seq samples with or without multiplexing for each breast cancer cell line were clustered together

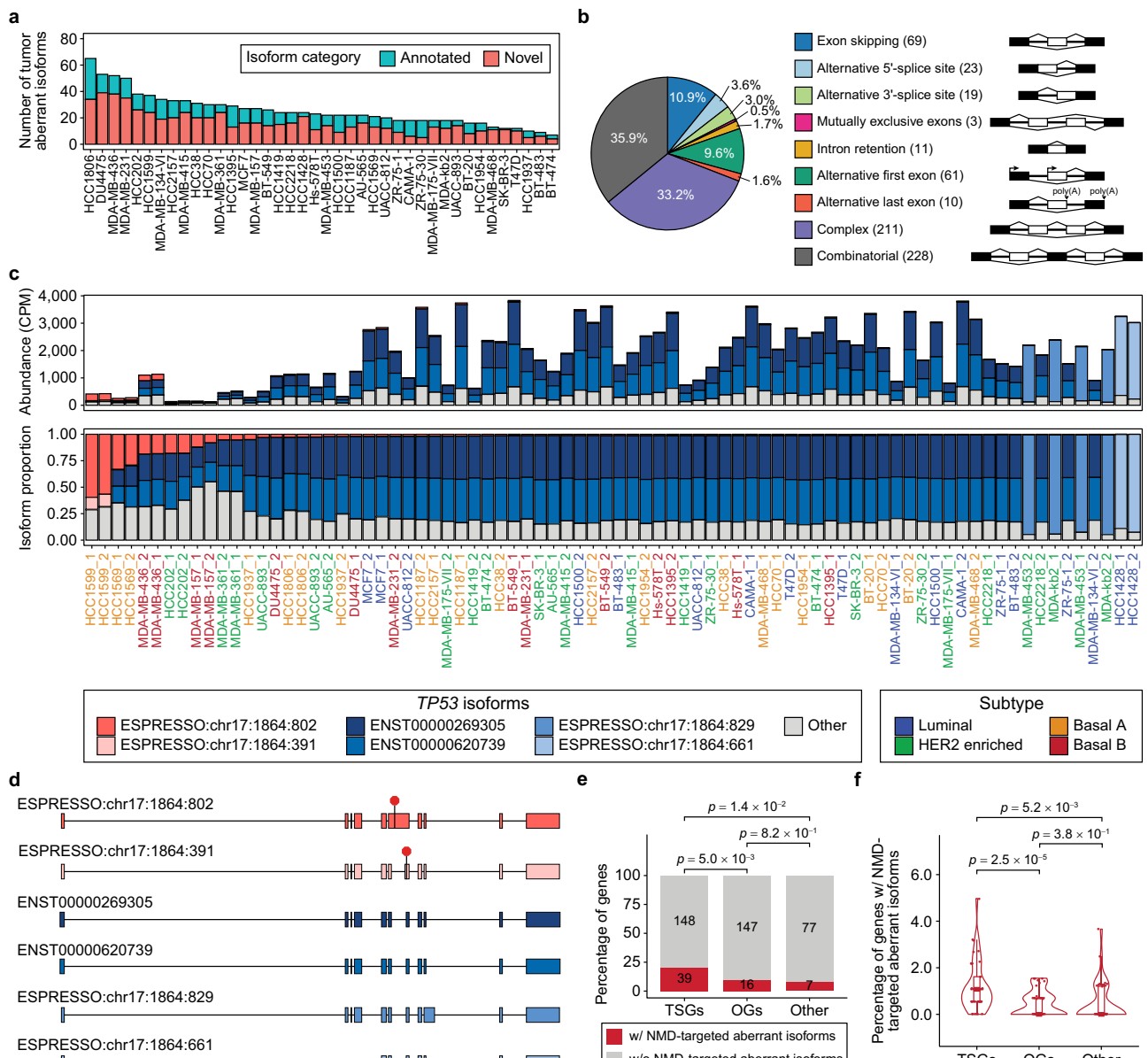

**Fig. 4 | Tumor aberrant transcript isoforms in 40 breast cancer cell lines.**
**a** Stacked barplots showing the number of annotated (turquoise) and novel (red) transcript isoforms identified as "tumor aberrant" across 40 breast cancer cell lines. **b** Categories of AS events in tumor aberrant transcript isoforms. Number in parenthesis: number of tumor aberrant transcript isoforms. **c** Stacked barplots showing estimated abundances (upper panel) and isoform proportions (lower panel) for *TP53* transcript isoforms discovered by TEQUILA-seq across 40 breast cancer cell lines. Red bars: isoforms of interest (ESPRESSO:chr17:1864:802, ESPRESSO:chr17:1864:391); navy bar: canonical isoform (ENST00000269305); lighter blue bars: three other most abundant *TP53* isoforms; gray bars: remaining

*TP53* isoforms. **d** Structures of *TP53* transcript isoforms. Boxes: exons. Line segments: introns. Red octagons: premature termination codons. **e** Stacked barplots showing percentages and numbers of genes with NMD-targeted tumor aberrant transcript isoforms within each gene category (TSG, OG, and Other) from the 468 actionable cancer genes. *P* values: two-sided Fisher's exact test. **f** Violin plots (median, interquartile range) with individual data points showing percentages of genes with NMD-targeted tumor aberrant transcript isoforms among all genes in each category (TSG, OG, and Other) detected in a given breast cancer cell line (mean value of two replicates per cell line, *n* = 40 in each violin plot). *P* values: two-sided paired Wilcoxon test. Source data are provided as a Source Data file.

based on transcript expression levels of 468 actionable cancer genes (Supplementary Fig. 13b). Overall, the capability for sample barcoding and multiplexing further increases the scalability and decreases the cost of TEQUILA-seq. It also allows TEQUILA-seq to harness the power of higher-throughput long-read sequencers (e.g., the ONT PromethION platform).

## Discussion

Targeted capture followed by long-read RNA-seq offers a powerful strategy for focused analyses of full-length transcript isoforms for preselected gene panels. It leverages the ability of long-read

sequencing platforms to sequence transcript molecules end-to-end, while circumventing their weaknesses of limited sequencing yield and low transcript coverage. Nevertheless, existing solutions for targeted long-read RNA-seq are either expensive[15], or difficult to set up and implement[16]. Here we present TEQUILA-seq, a broadly applicable method for targeted long-read RNA-seq. The TEQUILA process for synthesizing biotinylated capture oligos is versatile, easy to implement, and low-cost. Non-biotinylated oligo templates as starting material can be acquired as an array-synthesized oligo pool at a modest cost from various commercial vendors. By using nickase-triggered isothermal SDA, the TEQUILA process can generate large

quantities of biotinylated capture oligos from the limited starting material, enabling a large number of capture reactions, thus substantially reducing the per-reaction cost. As the nickase releases the synthesized strand from the universal adapter sequence, the TEQUILA probes are free of any artificial adapter sequence, with only complementary sequences against the targeted sequences. Importantly, using an input of only 2 ng of template oligo pool, we were able to generate up to 25 µg of TEQUILA probes (Supplementary Fig. 11), which can be used for 250 capture reactions, with 100 ng of probes used per capture reaction. Assuming that 2 ng of template oligo pool is regularly used per probe synthesis reaction, the total amount of template oligo pool (typically ranging from 50 ng to more than 200 ng based on our experience with commercial vendors) can be used for 25 to 100 TEQUILA probe synthesis reactions. Therefore, the total amount of TEQUILA probes can potentially be used for 6250 ($250 \times 25$) to 25,000 ($250 \times 100$) capture reactions. TEQUILA reduces the initial set up cost and drastically (by 2-3 orders of magnitude) reduces the per-reaction cost of targeted capture, as compared to a standard commercial solution, assuming a high number of re-uses (Supplementary Data 1, 2). With this cost structure, TEQUILA-seq can readily scale up to large cohorts with many biological samples.

We performed TEQUILA-seq of both synthetic spike-in RNA standards and human mRNAs, using multiple gene panels with sizes ranging from a small panel of 10 brain genes to a large panel of 468 actionable cancer genes. Our comprehensive benchmark analyses indicated consistently high on-target rate and fold enrichment across all samples and gene panels analyzed. Using synthetic spike-in RNA standards with known transcript structures and concentrations, we showed that TEQUILA-seq substantially improved the sensitivity of detecting low-abundance transcripts. At the same time, the estimated abundances of target transcripts based on TEQUILA-seq data correlated highly with the known spike-in concentrations (Fig. 2a). We also showed that TEQUILA-seq data did not exhibit length-dependent biases in transcript detection and quantification (Fig. 2b). Moreover, by comparing TEQUILA-seq data of a human gene panel to deep short-read RNA-seq data on the same sample, we showed that TEQUILA-seq preserved transcript isoform proportions of target genes (Fig. 2c). Overall, these results indicate that TEQUILA-seq provides a robust tool for transcript discovery and quantification for genes of interest.

Genomic DNA analysis of tumor samples, including both WGS and targeted sequencing, has been broadly used in research and clinical settings[30,31,56,57]. However, RNA-level dysregulation is prevalent in cancer transcriptomes[58], and recent studies have established the complementary utility of transcriptome sequencing for cancer genomic profiling[59–61]. By performing TEQUILA-seq of 468 actionable cancer genes across 40 breast cancer cell lines, we discovered numerous known or novel transcript isoforms with potential functional relevance. For example, we found that an alternative transcript isoform of *DNMT3B*, lacking two exons that encode part of its C-terminal catalytic domain, is highly enriched in basal B breast cancer cell lines (Fig. 3d, f). This finding has implications for the epigenetic regulation and DNA methylome of the basal B subtype, the most aggressive subtype of breast cancer[62,63]. We also discovered novel aberrant transcript isoforms of multiple genes encoding tumor suppressors, such as *TP53* and *RB1* (Fig. 4c, d and Supplementary Fig. 8). Using the full-length transcript information obtained by TEQUILA-seq, we can readily infer the consequences of isoform variation as it relates to transcript and protein products. For example, the aberrant transcript isoforms of *TP53* discovered in the HCC1599 cell line would introduce in-frame PTCs and trigger transcript degradation via the NMD pathway. Expanding this analysis to all aberrant transcript isoforms discovered in the breast cancer dataset, we found that TSGs are significantly more enriched for NMD-targeted aberrant transcript isoforms, as compared to OGs and other cancer genes (Fig. 4e, f). Thus, the TEQUILA-seq analysis reveals a common mechanism for inactivating TSGs in cancer cells, via aberrant alternative isoform variation coupled with transcript degradation via NMD.

TEQUILA-seq may facilitate many applications of targeted long-read RNA-seq. Here we illustrate a proof-of-concept application of TEQUILA-seq to actionable cancer genes; however, TEQUILA-seq can be applied to any gene panel of interest for focused discovery and quantification of transcript isoforms. For example, TEQUILA-seq of genes implicated in a given group of Mendelian disorders can be used for RNA-guided genetic diagnosis[64]. Likewise, TEQUILA-seq of genes involved in oncogenic gene fusions can be used for discovering actionable fusion transcripts towards precision oncology applications[65,66]. While the current TEQUILA-seq protocol based on the standard nanopore 1D cDNA sequencing workflow is designed for poly(A)+ RNAs, it can be easily adapted for non-poly(A)+ RNAs with minor modifications, such as poly(A)+ tailing of non-poly(A)+ RNAs. Importantly, TEQUILA probes can be used for targeted sequencing in both long-read and short-read RNA-seq workflows. One limitation of TEQUILA-seq, however, is that it is designed for cDNA sequencing but is not compatible with direct RNA-seq on the ONT platform for sequencing native RNA molecules[67]. Therefore, TEQUILA-seq cannot take advantage of the unique features of the ONT direct RNA-seq workflow, including its capability to directly read base modifications of RNA[10,68,69]. Beyond targeted RNA-seq, TEQUILA probes may also be used for various applications related to targeted DNA sequencing, such as targeted analysis of DNA methylation[70,71] and chromatin conformation[72,73]. Overall, hybridization capture-based enrichment is widely used for targeted DNA and RNA analysis[13,14]. We anticipate that the strategy for generating TEQUILA probes will have a broad impact, as the significant cost reduction, as well as ease of implementation, will pave the way for wide adoption in diverse biomedical and clinical settings.

## Methods
### Cell culture and treatment
SH-SY5Y human neuroblastoma cells (ATCC, #CRL-2266) were cultured in DMEM/F-12 (Gibco, #11330032) supplemented with 10% fetal bovine serum (FBS, Corning, #45000-734) and 100 U/ml penicillin-streptomycin (Gibco, #15140122). SH-SY5Y cells were maintained at 37 °C in a humidified chamber with 5% $CO_2$. The SH-SY5Y cell line was authenticated by short tandem repeat analysis and verified to be mycoplasma-free. A panel of 40 breast cancer cell lines (Supplementary Data 10) was purchased from the American Type Culture Collection (ATCC, Manassas, VA, USA 30-4500 K™). Cell lines were cultured according to ATCC recommendations (Supplementary Data 13) and were authenticated by the supplier. For NMD inhibition, approximately $1 \times 10^6$ of HCC1599 cells were treated with either DMSO (Sigma-Aldrich, #D2650) or a small-molecule NMD inhibitor SMG1i (compound 11j, MedChemExpress, #HY-124719) at the concentration of 300 nM for 4 h prior to harvest. Cells were pelleted at $300 \times g$ for 5 min and subject to RNA isolation or Western blotting.

### RNA extraction and preparation
The Spike-in RNA variants (SIRV) (SIRV-Set 4, Lexogen, #141.01) contain 69 SIRV isoforms (SIRV mix E0) and 15 long SIRV transcripts (long SIRVs) in equimolar ratios, as well as 92 non-isoform ERCC transcripts (ERCC mix) covering six orders of magnitude of concentration range. Human brain total RNA (50 µg, Clontech, Cat. #636530, Lot #2006022) was isolated from pooled tissues of multiple donors, as described by the manufacturer. Total RNA was extracted from the SH-SY5Y cell line and 40 breast cancer cell lines using TRIzol reagent (Invitrogen, #15596018). RNA concentrations and RNA integrity were measured with a NanoDrop 2000 Spectrophotometer and Agilent 4200 TapeStation, respectively.

### RT-PCR validation and Sanger sequencing of cDNA

Total RNA was treated with RNase-free DNase I by using the TURBO DNA-free Kit (Invitrogen, #AM1907). The cDNA was synthesized from 1 μg of total RNA using oligo (dT)15 primed reverse transcription as described in the Maxima H minus reverse transcriptase (Thermo Scientific, #EP0751) protocol. Next, PCR was performed in a 20-μl reaction by using first-strand cDNA synthesized from 50 ng of total RNA, 10 μl of KAPA HiFi ReadyMix (KAPA Biosystems, #KK2602), and 10 pmol of a primer pair. All primer pairs are listed in Supplementary Data 14. PCR amplification was carried out in a Veriti 96-well Thermal Cycler (Applied Biosystems, #43-757-86) by incubating the mixture at 95 °C for 3 min, followed by 26 cycles of (98 °C for 20 s, 65 °C for 20 s, and 72 °C for 45 s) with a final extension at 72 °C for 2 min. Amplified products were analyzed by electrophoresis in 2% agarose gels and a D1000 ScreenTape assay on an Agilent 4200 TapeStation. Splice junction sequences of transcript isoforms were confirmed by Sanger sequencing of the DNA amplicons, which were separated by DNA electrophoresis. Gel extraction was performed using the QIAquick Gel Extraction Kit (Qiagen, #28706X4).

### Genomic DNA isolation and Sanger sequencing

Genomic DNA was isolated using TRIzol reagent (Invitrogen) according to the DNA isolation protocol from the manufacturer. DNA concentration and integrity were measured by a NanoDrop 2000 Spectrophotometer and Genomic DNA ScreenTape assay on an Agilent 4200 TapeStation, respectively. PCR was performed in a 50-μl volume using 50 ng of genomic DNA, 25 μl of KAPA HiFi ReadyMix, and 20 pmol of a primer pair. All primer pairs are listed in Supplementary Data 14. PCR amplification was carried out in a Veriti 96-well Thermal Cycler by incubating the mixture at 95 °C for 3 min, followed by 30 cycles of (98 °C for 20 s, 65 °C for 20 s, and 72 °C for 1 min) with a final extension at 72 °C for 2 min. Amplified products were separated by electrophoresis in 1.5% agarose gels, and bands were purified with QIAquick Gel Extraction Kit. Sequences of purified DNA amplicons were confirmed using Sanger sequencing with the same primers used in PCR.

### Western blotting

Approximately $1 \times 10^6$ of HCC1599 cells treated with either DMSO or SMG1i (300 nM, 4 h) were pelleted at $300 \times g$ for 5 min and resuspended in 200 μl of ice-cold RIPA lysis and extraction buffer (Thermo Scientific, #89901) containing protease inhibitors (Pierce, #A32955). The samples were gently pipetted ten times to lyse and left on ice for 10 min. The lysates were centrifuged at $20,000 \times g$ for 5 min and the supernatant was transferred to a new tube. Samples were then added with 40 μl of SDS sample loading buffer (375 mM Tris-HCl, 9% SDS (w/v), 50% glycerol (v/v), 9% 2-mercaptoethanol (v/v), 0.03% bromophenol blue (w/v)), and boiled at 95 °C for 5 min. Afterward, 10 μl of boiled samples were electrophoresed in a 10% mini-PROTEAN TGX precast protein gel (Bio-Rad, #4561035), and transferred to a polyvinylidene difluoride (PVDF) membrane using TransBlot Turbo transfer system (Bio-Rad, #170-4272). The membranes were blocked with 5% non-fat milk in 1× TBST (Cell Signaling Technology, #9997) at room temperature for 1 h and probed at 4 °C overnight with a 1:1000 dilution of Phospho-Upf1 (Ser1127) antibody (Sigma-Aldrich, #07-1016) and a 1:20,000 dilution of β-actin loading control monoclonal antibody (BA3R) (Invitrogen, #MA5-15739). The blots were then incubated at room temperature for 1 h with HRP-conjugated goat anti-rabbit IgG (1:2000 dilution, Agilent Technologies, #P044801-2) and goat anti-mouse IgG secondary antibody (1:5000 dilution, Invitrogen, #PI32430), respectively. Finally, the blots were detected via chemiluminescence with SuperSignal West Femto maximum sensitivity chemiluminescent substrate (Thermo Scientific, #34094) and imaged on a ChemiDoc XRS+ system with Image Lab software (Bio-Rad, #1708265).

### Short-read RNA-seq library preparation and sequencing

Short-read sequencing libraries were prepared using 1 μg of total RNA extracted from SH-SY5Y cells and spiked-in with 25 pg of SIRV-Set 4 RNA, following the TruSeq Stranded mRNA protocol (Illumina, #20020595). All short-read libraries ($n = 3$ replicates) were sequenced on an Illumina NovaSeq 6000 sequencer with 150-bp paired-end sequencing, according to the manufacturer's protocol.

### Direct RNA library construction and nanopore sequencing

A 20-μg aliquot of total RNA was subjected to poly(A)+ RNA selection using the Dynabeads mRNA DIRECT purification kit (Invitrogen, #61011) following the manufacturer's instructions. Approximately 500 ng of the resulting poly(A)+ RNA, along with 5 ng of SIRVs, were pooled as input for direct RNA library generation. Libraries were made by following the standard ONT SQK-RNA002 protocol with the optional reverse transcription step included. All libraries were loaded onto R9.4.1 flow cells and sequenced on MinION/GridION devices (ONT, Oxford, UK). Sequencing summary statistics, including library information, chemistry, and sequencing output, are detailed in Supplementary Data 15.

### Full-length cDNA synthesis

A 200-ng aliquot of total RNA, together with 5 pg of SIRV-Set 4 RNA, were used as templates for cDNA synthesis. Briefly, the reverse transcription and template-switching reaction was performed by using Maxima H minus reverse transcriptase under the following conditions: 42 °C for 90 min, followed by 85 °C for 5 min. First-strand cDNA was amplified by PCR with KAPA HiFi ReadyMix by incubating the mixture at 95 °C for 3 min, followed by 11 cycles of (98 °C for 20 s, 67 °C for 20 s, and 72 °C for 5 min) with a final extension at 72 °C for 8 min. PCR products were purified using 0.8× volumes of SPRIselect beads (Beckman Coulter, #B23318). Amplified cDNA was measured using the Qubit dsDNA High Sensitivity assay and Agilent High Sensitivity D5000 ScreenTape assay on a 4200 TapeStation. Sequences of oligos/primers are listed in Supplementary Data 14.

### 1D library construction and nanopore sequencing

Nanopore 1D libraries were constructed using 1 μg of amplified cDNA according to the standard ONT SQK-LSK109 protocol. Briefly, cDNA products were end-repaired and dA-tailed using NEBNext Ultra II End Repair/dA-Tailing Module (NEB, #E7546) by incubating at 20 °C for 20 min and 65 °C for 20 min. The cDNA was then purified with 0.7× volumes of SPRIselect beads and eluted in 60 μl of nuclease-free water. Adapter ligation was performed using NEBNext Quick T4 DNA ligase (NEB, #E6056) at room temperature for 10 min. After ligation, libraries were purified using 0.45× volumes of AMPure XP beads (Beckman Coulter, #A63881) and short fragment buffer. The final libraries were loaded onto R9.4.1 flow cells and sequenced on MinION/GridION devices. Sequencing summary statistics, including library information, chemistry, flow cell types, and sequencing output, are detailed in Supplementary Data 15.

### Capture probe synthesis

IDT Lockdown probes (Integrated DNA Technologies) were designed and synthesized for a test panel of 10 human genes with long transcripts in the brain, including *DAB1, DLG4, GRIN1, HTT, LRP8, MAPT, NRXN1, NUMB, RBFOX1*, and *SCN8A*. The probes are 120-nt long oligos that are biotinylated at their 5′ ends. Probes were designed to tile across all annotated exons, including UTRs, of test panel genes with 1× tiling density (Supplementary Data 14).

TEQUILA probes were synthesized in two steps. First, Twist oligo pools (Twist Bioscience) were designed and synthesized for three custom-designed gene panels, which are detailed in Supplementary Data 14. The oligos are 150-nt long and contain a 3′ end 30-nt universal primer binding sequence (5′-CGAAGAGCCCTATAGTGAGTCGT

ATTAGAA-3'). The remaining 120 nt were designed to tile across all annotated exons, including UTRs, of target genes with 1× tiling density, by using the xGen Hyb Panel Design Tool (https://www.idtdna.com/pages/tools/xgen-hyb-panel-design-tool). Specifically, BED/FASTA files, gene symbols, or transcript accession numbers of target genes were uploaded as input, and "Homo sapiens (Human) NCBI GRCh37.p13 (hg19)" was used as the reference. Probe length and probe tiling density were set to 120 nt and 1×, respectively. A spreadsheet of 120-nt oligo sequences was generated as output along with information on target base counts, probe counts, and coverage percentage. Next, oligo pools were amplified and biotin-labeled using nickase-induced linear SDA. Briefly, a 40 µl reaction was assembled on ice; it contained 2–10 ng of the oligo pool as ssDNA templates, 5 µl of 10× NEBuffer 3.1, 2 mM DTT, 0.25 µM RC-oligo (5'-TTCTAA-TACGACTCACTATAGGGCTCTTCG-3'), 0.4 mM dTTP, 0.6 mM dATP, 0.6 mM dCTP, 0.6 mM dGTP, and 0.2 mM biotin-16-aminoallyl-2'-dUTP (TriLink BioTechnologies, #N-5001). The mixture was incubated at 95 °C for 2 min and then ramped down to 4 °C at a rate of 0.1 °C/s. Initial strand extension of primers was performed at 37 °C for 10 min using 5 µM of ssDNA binding protein (T4 Gene 32 Protein, NEB, #M0300S) and 0.8 U/µl of Klenow Fragment (3'–5' exo-) DNA polymerase (NEB, #M0212M). Nickase-induced linear SDA was then performed at 37 °C for 4–16 h using 3 nM (0.04 U/µl) of Nt.BspQI (NEB, #R0644S). Synthesized probes were purified with 1.8× volumes of AMPure XP beads and quantified by NanoDrop 2000 Spectrophotometer.

## Hybridization and capture

All hybridization and capture experiments were done following a protocol from IDT ("Hybridization capture of DNA libraries using xGen Lockdown probes and reagents"). Briefly, approximately 500 ng of amplified cDNA were denatured at 95 °C for 10 min and then incubated with either 3 pmol of IDT xGen Lockdown probes or 100 ng of TEQUILA probes at 65 °C for 12 h. Next, 50 µl of M-270 streptavidin beads (Invitrogen, #65306) were added to the mixture, which was incubated at 65 °C for 45 min. The mixture was then immediately subjected to a series of high-temperature and room temperature washes, according to the IDT xGen Lockdown protocol. The resulting bead solution was resuspended in 40 µl of TE buffer (Invitrogen, #12090015). For hybridization with the addition of blocking oligonucleotides, prior to denaturing step, approximately 500 ng of amplified cDNA were combined with blocking oligos, including oligos targeting the adapters used in probe synthesis (1 nmol of CGAAGAGCCCTA-TAGTGAGTCGTATTAG-/3SpC3/, IDT), oligos targeting the primers for cDNA amplification (1 nmol of AAGCAGTGGTATCAACGCAGAGT, IDT), and oligo(dT)18 with a three-carbon spacer at the 3' end targeting the poly(A) tail (1 nmol of TTTTTTTTTTTTTTTTTT/3SpC3/, IDT).

## Post-capture amplification and nanopore sequencing

On-bead PCR was performed for the streptavidin bead-captured cDNA using KAPA HiFi ReadyMix by incubating at 95 °C for 3 min, followed by 12 cycles of (98 °C for 20 s, 67 °C for 20 s, 72 °C for 5 min), with a final extension at 72 °C for 8 min. PCR products were purified using 0.7× volumes of SPRIselect beads, and measured using the Qubit dsDNA High Sensitivity assay and Agilent High Sensitivity D5000 ScreenTape assay on a 4200 TapeStation. Amplified cDNA was subjected to library construction and nanopore sequencing. Sequencing summary statistics, including library information, chemistry, and sequencing output, are detailed in Supplementary Data 15.

For TEQUILA-seq of 468 actionable cancer genes across 40 breast cancer cell lines, the two replicates of TEQUILA-seq libraries for each breast cancer cell line were sequentially sequenced on one R9.4.1 MinION flow cell. The flow cell wash kit (ONT, EXP-WSH004) was used to wash out the first library and then refresh the flow cell for the second

library to be loaded and sequenced. All fluids from the waste channel were removed through waste port 1. Next, a mix of 2 µl of wash mix (WMX, containing DNase I) and 398 µl of wash diluent (DIL) was loaded into the flow cell priming port and incubated for 90 min. After digestion, all fluids from the waste channel were removed through waste port 1. A second library was then loaded for sequencing by following the instructions in the "Priming and loading the flow cell" section of the 1D ligation kit protocol (SQK-LSK109).

For barcoding and multiplexing of TEQUILA-seq libraries, approximately 200 ng of post-capture amplified cDNA from each sample were used according to the ONT SQK-NBD114.24 protocol. Briefly, cDNA products were end-repaired and dA-tailed using NEB-Next Ultra II End Repair/dA-Tailing Module by incubating at 20 °C for 20 min and 65 °C for 20 min. The cDNA was then purified with 1× volume of AMPure XP beads and eluted in 10 µl of nuclease-free water. Native barcode ligation was performed using 7.5 µl of end-prepped DNA, 2.5 µl of native barcodes, and 10 µl of Blunt/TA Ligase Master Mix (NEB, #M0367) by incubating at room temperature for 20 min. To inactivate the ligase, 2 µl of EDTA was added to each ligation reaction. Barcoded cDNA samples were then pooled and purified using 0.4× volume of AMPure XP beads and eluted in 35 µl of nuclease-free water. Next, adapter ligation was performed using 30 µl of pooled barcoded samples, 5 µl of native adapter, 10 µl of 5× NEBNext Quick ligation reaction buffer, and 5 µl of NEBNext Quick T4 DNA ligase at room temperature for 20 min. After ligation, libraries were purified using 0.45× volume of AMPure XP beads and short fragment buffer. The final libraries were loaded onto R10.4.1 flow cells and sequenced on MinION/GridION devices.

## Basecalling and alignment of nanopore sequencing data

Basecalling of raw nanopore data was performed in fast mode using Guppy (v4.0.15) with the following settings: '*guppy_basecaller --input_path raw_data --save_path output_folder --config corresponding config file*' (https://community.nanoporetech.com/downloads). Basecalling of whole-transcriptome 1D cDNA sequencing and TEQUILA-seq data was done using config file '*dna_r9.4.1_450bps_fast.cfg*', and basecalling of direct RNA sequencing data was done using config file '*rna_r9.4.1_70bps_fast.cfg*'. For raw nanopore data described in the section "Technical refinements of TEQUILA-seq", basecalling was performed in super-accurate mode using Guppy (v6.4.2) with the '*dna_r9.4.1_450bps_sup.cfg*' config file for R9.4.1 and the '*dna_r10.4.1_-e8.2_260bps_sup.cfg*' config file for R10.4.1 chemistry. Basecalled reads from multiplexed nanopore sequencing data were also demultiplexed using Guppy (v6.4.2) with the parameter: '*--barcode_kits SQK-NBD114-24*'.

Basecalled reads were mapped to either the GRCh37/hg19 reference genome or SIRV genome from Lexogen (SIRV-Set 4) using mini-map2 (v2.17) with parameters: '*-a -x splice -ub -k 14 -w 4 --secondary=no*'. Specifically, we provided minimap2 with transcript annotations of GENCODE v34 (https://www.gencodegenes.org/human/release_34lift37.html) when mapping reads to the GRCh37/hg19 reference genome. We provided minimap2 with SIRV-Set 4 transcript annotations when mapping reads to the SIRV genome.

## Discovery and quantification of transcript isoforms

Full-length transcript isoforms were discovered and quantified from long-read alignment files using ESPRESSO (v1.2.2) with default settings. ESPRESSO is a computational tool we developed for the robust discovery and quantification of transcript isoforms from error-prone long-read RNA-seq data[17] (https://github.com/Xinglab/espresso). Specifically, ESPRESSO was applied to the following sets of nanopore RNA-seq data:

1. Whole-transcriptome 1D cDNA sequencing data and targeted sequencing data (IDT probes or TEQUILA probes) of 10 human brain genes (Supplementary Data 3) on human brain cDNA samples (*n* = 3 replicates per sequencing protocol).

2. Direct RNA sequencing data, whole-transcriptome 1D cDNA sequencing data, and TEQUILA-seq data (4, 8, and 48 h of sequencing time) of a panel of 46 ERCC transcripts and five long SIRV transcripts (Supplementary Data 5), as well as 221 genes encoding splicing factors (Supplementary Data 7) on SH-SY5Y cells ($n = 3$ replicates per sequencing protocol).

3. TEQUILA-seq data of 468 actionable cancer genes (Supplementary Data 9) on 40 breast cancer cell lines ($n = 2$ replicates per cell line) (Supplementary Data 10).

4. Whole-transcriptome 1D cDNA sequencing data on four breast cancer cell lines: HCC1806, MDA-MB-157, AU-565, and MCF7 ($n = 1$ per cell line).

5. TEQUILA-seq data of 468 actionable cancer genes on four breast cancer cell lines: HCC1806, MDA-MB-157, AU-565, and MCF7 without blocking oligos (without multiplexing) or with blocking oligos (without multiplexing or with multiplexing) (described in the section "Technical refinements of TEQUILA-seq").

Estimated read counts for all transcript isoforms discovered in a sample (i.e., those with a nonzero read count) were normalized into counts per million (CPM) by dividing the number of reads assigned to a transcript isoform by the total number of reads mapped to the reference genome, and multiplying this number by one million. The proportion of a transcript isoform was calculated by dividing the CPM value of that transcript isoform by the CPM value of the corresponding gene (i.e., the sum of CPM values over all transcript isoforms discovered for the gene).

### Calculation of on-target rate and fold enrichment
For each sample, we calculated an on-target rate by dividing the number of RNA-seq reads mapped to target genes (with a mapping quality score ≥1) by the total number of reads mapped to the reference genome (with a mapping quality score ≥1). Fold enrichment on a given sample was calculated by dividing the on-target rate for targeted RNA-seq (TEQUILA-seq or xGen Lockdown-seq) by the on-target rate for non-capture control.

### Quantification of exon skipping events using short- and long-read RNA-seq data
We aligned short-read RNA-seq data to the GRCh37/hg19 reference genome using STAR (v2.6.1d) in two-pass mode with default settings and transcript annotations of GENCODE v34 (https://www.gencodegenes.org/human/release_34lift37.html). Exon skipping events were discovered and quantified (as percent spliced in, ψ) from short-read alignment files using rMATS (v4.1.1) with default settings[29].

We compiled high-confidence exon skipping events from short-read RNA-seq data based on the following criteria: (1) the average number of short reads supporting the two exon-included splice junctions or the number of short reads supporting the exon-skipped splice junction is ≥10, (2) the ratio between the numbers of short reads supporting each of the two exon-included splice junctions is between 0.2 and 5, (3) the short-read ψ value is between 0.01 and 0.99, and (4) none of the four splice sites associated with the exon skipping event (i.e., the two splice sites of the upstream intron and the two splice sites of the downstream intron) is involved in other AS events discovered from short-read RNA-seq data.

For each exon skipping event discovered from short-read data, we also computed ψ values based on long-read data using the following equation:

$$\psi = \frac{I}{I + S}$$

where $I$ is the sum of CPM values for transcript isoforms carrying both of the exon-included splice junctions, and $S$ is the sum of CPM values for transcript isoforms carrying the exon-skipped splice junction.

### Identification of tumor subtype-associated transcript isoforms
We identified tumor subtype-associated transcript isoforms using a panel of 40 breast cancer cell lines. For each subtype (luminal, HER2 enriched, basal A, or basal B), we used a two-sided Student's $t$-test to compare the proportion of a transcript isoform between cell lines belonging to the given subtype and all other cell lines. We defined tumor subtype-associated transcript isoforms as those meeting the following criteria: (1) FDR-adjusted $p$ value ≤5% based on Benjamini–Hochberg correction and (2) the mean transcript isoform proportion across cell lines of the given subtype is greater than the mean transcript isoform proportion over all other cell lines by at least 10% (Supplementary Data 12).

### Identification of tumor aberrant transcript isoforms
We defined "tumor aberrant" transcript isoforms as transcript isoforms that are present at significantly elevated proportions in at least one but no more than 4 (i.e., ≤10%) breast cancer cell lines by using the following statistical procedure:

For each gene, we generated an $m$-by-80 matrix composed of read counts (rounded to the nearest integer) for $m$ discovered transcript isoforms across 80 TEQUILA-seq samples (two replicates for each of the 40 breast cancer cell lines). Using this matrix, we computed the total gene expression level in each sample as the sum of read counts over all transcript isoforms of the gene. We ignored genes that only had one discovered transcript isoform or were only expressed in a single sample. We also omitted samples from the matrix if the given gene was not expressed in those samples.

Next, we ran a chi-square test of homogeneity (FDR <1%) on the matrix to assess whether transcript isoform proportions for the given gene are homogenous across the considered samples. Focusing on genes identified by the chi-square test with FDR <1%, we ran a post hoc test to identify sample-isoform pairs in which the transcript isoform proportion in the given sample is significantly higher than the overall transcript isoform proportion over all samples (i.e., the sum of read counts of the transcript isoform over all samples divided by the sum of read counts of the gene over all samples) (one-tailed binomial test, FDR <1%). Using transcript isoforms identified by this post hoc test, we next identified cell line-isoform pairs for which the transcript isoform shows significantly elevated usage in a given cell line. Specifically, these pairs were required to meet the following criteria: (1) the transcript isoform has an adjusted $p$ value <1% (post hoc test) using Benjamini–Hochberg correction for both replicates of the given cell line and (2) the transcript isoform proportions in both replicates are ≥10% higher than the overall transcript isoform proportion over all samples.

Finally, we defined tumor aberrant transcript isoforms based on the following criteria: (1) the transcript isoform shows significantly elevated usage in at least one but no more than 4 cell lines (i.e., ≤10% of our breast cancer cell line panel) and (2) the transcript isoform is not the canonical transcript isoform of the corresponding gene, as defined by the Ensembl database (Release 100, April 2020).

### Classification of AS events underlying tumor aberrant transcript isoforms
To characterize RNA processing changes associated with tumor aberrant transcript isoforms, we compared the structure of each tumor aberrant transcript isoform with the structure of the canonical transcript isoform of the corresponding gene. Local differences in transcript structure were classified into seven basic AS categories[8], including (1) exon skipping, (2) alternative 5'-splice site, (3) alternative 3'-splice site, (4) mutually exclusive exons, (5) intron retention, (6) alternative first exon, and (7) alternative last exon. Any local differences in transcript structure that could not be classified as one of the seven basic AS categories were classified as "complex". If a tumor aberrant transcript isoform had more than one AS event relative to the canonical transcript isoform, it was labeled as "combinatorial".

## Identification of NMD-targeted transcript isoforms

All transcript isoforms discovered by ESPRESSO were classified into the following three categories: (1) transcripts annotated in GENCODE (v34lift37) as "basic" (i.e., full-length) protein-coding or NMD-targeted, (2) transcripts annotated in GENCODE but not labeled as "basic" protein-coding or NMD-targeted, (3) novel transcripts discovered by ESPRESSO. For transcripts classified into categories (2) or (3), we retrieved their sequences from the GRCh37/hg19 reference genome and searched for ORFs. Specifically, we used the longest ORF for a given transcript and required it to encode at least 20 amino acids. Among transcripts with predicted ORFs, we identified those that may be NMD-targeted using the following criteria[45]: (1) the transcript is ≥200 nt long, (2) the transcript contains at least one splice junction, and (3) the predicted stop codon is ≥50 nt upstream of the last splice junction (i.e., the transcript harbors a PTC).

## Reporting summary

Further information on research design is available in the Nature Portfolio Reporting Summary linked to this article.

## Data availability

Raw and processed data from Illumina short-read RNA-seq and nanopore long-read RNA-seq were uploaded to GEO under accession number GSE213984. Source data are provided with this paper.

## Code availability

The scripts used for processing, analyzing, and visualizing TEQUILA-seq data are publicly available on GitHub (https://github.com/Xinglab/TEQUILA-seq) and deposited in Zenodo (https://doi.org/10.5281/zenodo.8018742).

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

## Acknowledgements

This work was supported by National Institutes of Health grants R01GM121827 (L.L.), R01GM088342, R56HG012310, and U01CA233074 (Y.Xing), and T32HG000046 (R.W.). We thank Yuan Gao, Liling Wan, and Wanding Zhou for their helpful discussions.

## Author contributions

L.L. and Y.Xing conceived the study and supervised the research. L.L., Y.Xing, and F.W. designed the research and developed the methodology. Y.Xu, R.W., and E.K. developed analytic tools. F.W., N.S., A.N., and S.G. performed the experiments and generated the data. Y.Xu, R.W., and B.Z. analyzed the data. F.W., Y.Xu, R.W., K.E.K.-E., Y.Xing, and L.L. wrote the paper with input from all other authors.

## Competing interests

Y.Xing is a scientific co-founder of Panorama Medicine. L.L., Y.Xing, and F.W. are named inventors on a patent application (application number: PCT/US22/79537, in process) filed by The Children's Hospital of Philadelphia covering the TEQUILA method. The other authors declare no competing interests.
