## [Peer Review File · Nature Communications]

TEQUILA-seq: A versatile and low-cost method for targeted long-read RNA sequencingREVIEWER COMMENTS

Reviewer #1 (Remarks to the Author):

In their paper “TEQUILA-seq: A versatile and low-cost method for targeted long-read RNA sequencing”, Wang et al. introduce the a method for generating low cost enrichment probes using nicking-
endonuclease (nickase)-triggered isothermal strand displacement amplification (SDA). They benchmark
the use of these probes by enriching full-length cDNA samples.

First, the authors compare TEQUILA probes to IDT xGen probes on a panel of 10 genes which they
enriched from a commercially available sample of human brain RNA. Next they test whether enrichment
affects relative quantification using spike-in controls. Further they enriched a panel of cancer genes
from cDNA of 40 breast cancer cell lines and performed isoform-level analysis on the resulting data
using their new “Espresso” analysis pipeline.

I think the easy generation of cheap enrichment probes as described by the manuscript is a great
addition to the field. This is something I would personally try and use and I believe there exists a real use
case for this. The fact that these probes seem to perform almost identically to xGen probes is
remarkable.

Using full-length cDNA analysis for the benchmarking of these probes is interesting because that
represents a still somewhat niche application of target enrichment. Also, analyzing full-length cDNA
comes with the added challenge of isoform identification and quantification which can make it difficult
to disentangle the performance of the enrichment probes and the cDNA sequencing and analysis
approach.

There are some really cool things in the manuscript, like explaining alternative isoform expression with
structural variants in cell lines and overall, I think this is a strong manuscript. I do have some points that I
would like to see addressed or considered:

1) Because there is a considerable focus on the cost-effectiveness of the probes, I would like to see more
detail on the process of probe generation:

A) There is some information on yields in Supplemental Figure 1 but I would like to see actual data on
the yields of probe generation reactions. What is the range of material individual reactions produce? I
think having that information in a table or figure would be very useful.

B) Size of the generated probes: Do the sizes of the generated probes match the length of the template
oligos? Gel or tapestation images would be great for this.

2) Enrichment reactions usually use some form of blocking oligos. However the manuscript does not
mention those. Were the enrichment reactions indeed performed without blocking oligos?

3) Does enrichment affect read lengths? The methods section show different SPRI bead ratios are used

after initial cDNA synthesis (0.8) and after PCR following enrichment (0.7). What is the reason for and result of this?

4) Often many samples can be multiplexed during target enrichment to minimize costs but also batch effects of enrichment and sequencing. Why weren't samples in this study indexed and pooled before enrichment? Even R9.4/guppy4 should be accurate enough to index full-length cDNA with for example 20nt sample barcodes in oligodT primers. In the unlikely case that pooling samples before enrichment using TEQUILA probes isn't possible, this should be mentioned when cost per sample is discussed.

5) cDNA sequencing on the MinION is notorious for batch effects. Read numbers can vary greatly (and do in this study according to Table S15) and read length distributions can be all over the place. This can affect isoform analysis. Further, it would affect the quantification of transcripts. This leads me to a two questions:

What are the read length distributions for all experiments in this manuscript? Violin plots would be nice but a table with Median and 25th and 75th percentile would do.

If read lengths are different between experiments I would like to see a discussion how this is going to affect isoform analysis.

6) The quantification of the 5 enriched long SIRVs is interesting for several reasons. First, what 5 SIRVs were selected and what is their respective length. How does this compare to the read length of the entire sequencing run (before and after enrichment) and more specifically, how does this relate to the lengths of the reads aligning to each SIRV? If the length of SIRVs and its aligning reads is not in agreement, how exactly is quantification performed by ESPRESSO?

Long SIRVs are an interesting example because they all have entirely unique sequences meaning transcript length could be irrelevant to their quantification (i.e. sequencing only the most 3' 50nt of them would be enough to identify them - an edgecase quite different from comparing isoforms within a gene).

Reviewer #2 (Remarks to the Author):

Long-read sequencing platforms, such as Pacific Biosciences and Oxford Nanopore Technologies are powerful and enabling technologies that allows for full-length transcript sequencing. Despite their potentials in a variety of biomedical and clinical applications, one major bottleneck of the long-read sequencing platforms is that their throughput is several orders of magnitude lower than that of short-read platforms. One way to overcome this bottleneck is through targeted sequencing for predefined gene panels. However, existing solutions for targeted long-read RNA-seq are either expensive, or difficult to implement. In the current study, Wang et al. describes the development of TEQUILA-seq, a versatile, easy-to-implement, and low-cost method for targeted long-read RNA-seq. The major innovation of TEQUILA-seq is that it uses nicking-endonuclease- triggered isothermal strand displacement amplification to synthesize large quantities of biotinylated capture oligos from an array-

synthesized pool of non-biotinylated oligo templates. As a result, TEQUILA-seq reduces the per-reaction cost of targeted capture by 2-3 orders of magnitude, in comparison with the standard commercial solution. They further applied TEQUILA-seq to a panel of 468 actional cancer genes across representative breast cancer cell lines and identified transcript isoforms enriched in specific subtypes and discovered novel transcript isoforms in extensively studied cancer genes. TEQUILA-seq is a long-awaited method that will greatly facilitate the wide adoption of targeted long-read RNA-seq in different basic research and clinical application settings. My detailed comments are listed as follows.

1. The key innovation of TEQUILA-seq is a new low-cost method for generating large quantities of biotinylated capture oligos. This innovation will potentially significantly reduce the cost of targeted DNA-seq as well. The authors only briefly mentioned the adoption of this method for targeted DNA-seq in the discussion part, which is not very visible. It would be worth expanding on the potential impact of this new biotinylated capture oligo generation method in both introduction and discussion section, on targeted long/short-read DNA-seq for population-scale genetics study and precision medicine.

2. The ERCC standard synthetic transcripts of unique sequences and diverse concentrations were used to evaluate TEQUILA-seq's capability of preserving and quantification of target transcripts. The use of synthetic transcript pool alone for evaluation has limitations because it does not reflect the complexity of the human transcriptome in cells/tissues. Therefore, it is important to evaluate TEQUILA-seq's performance of preserving and quantification of target transcripts by using the synthetic transcripts as the spike-in that is mixed with the RNAs extracted human cells/tissues.

3. The SIRVs comprising 15 synthetic transcripts of equimolar concentrations that cover transcript lengths from 4,000 to 12,000 nt was used to determine whether TEQUILA-seq data exhibit any transcript length-dependent biases. To justify the evaluation using SIRVs, further information needs to be provided. For example, what is the length distribution of isoforms for typical human genes and how representative the length distribution of SIRVs isoforms is among human genes? How far we can extrapolate the results from SIRV to other human genes? It is possible that the option of synthetic isoforms for this type of evaluation is limited. It is thus important to discuss about the limitation of using SIRVs for evaluation.

4. In the overview of TEQUILA-seq, it says "full-length cDNAs are synthesized from poly(A)+ RNAs by reverse transcription and PCR amplification". As there are many non-polyA transcripts in the human transcriptome, it would be important to describe the situation for non-poly (A)+ RNAs, even if the current study focuses on poly(A)+ ones.

5. By using TEQUILA-seq, the authors identified transcript isoforms enriched in specific subtypes and discovered novel transcript isoforms in extensively studied cancer genes in breast cancer cell lines. As the cell line could be from tumors, how many of these discoveries have supporting evidence from RNA-seq data generated in human tumors, e.g., from TCGA breast cancer data?

6. In the method section, it is not very clear how the probe design was performed. What

methods/software were used?

7. It says “We defined tumor aberrant transcript 245 isoforms as transcript isoforms that are present at significantly elevated proportions in at least one 246 but no more than 4 (i.e. $\leq 10\%$) breast cancer cell lines”. It is not clear why these isoforms are defined as “tumor aberrant transcript” because there was no comparison between tumor cells and normal cells here and these transcripts could show high expression in normal cells/tissues as well. Because these aberrantly expressed isoforms only occurs in $<10\%$ breast cancer cell lines, cautions need to be taken about whether they are true aberrantly expressed isoforms or they were identified due to some technical artifacts from sequencing data. Are these transcripts associated/potentially caused by some specific mutations in the small proportion of cell lines?

8. Is there any literature knowledge about the biological functions of the identified subtype specific isoforms?

9. It was found that many tumor aberrant transcript isoforms identified by TEQUILA-seq contain premature termination codons, which would target the transcript isoform for degradation via nonsense-mediated (NMD) mRNA decay. It remains unclear why the NMD-targeted isoforms show higher expression than other isoforms that are not NMD-targeted. Do these genes with dominant NMD-targeted isoforms show lower overall expression than the same genes with less NMD-targeted isoforms in other cell lines? It would be important to clarify and discuss the related issues.

10. The current study relies on the software ESPRESSO. Has it been peer-reviewed?

Reviewer #3 (Remarks to the Author):

In this paper a new methodology called TEQUILA-seq is presented which aims to reduce the cost of targeted RNA enrichment through the reuse of oligo pools. The authors comprehensively demonstrate that their methodology is able to greatly enrich required targets and that quantification information about those targets is not lost during the process. This is somewhat surprising given the bias that is sometimes seen in some of these studies. That aside, the experiments they perform include a mixture of real and synthetic samples and the data presented backs up their claims. Furthermore, they are able to apply the approach to 40 breast cancer cells lines and efficiently identify interesting transcript iso-forms enriched in key subtypes.

Overall, this paper is excellently written with a very clear narrative that addresses all the core questions you might have with such an approach. In terms of novelty, the use of enrichment probes is not new, and the methodology chosen for that element is near identical to commercially available methods (e.g. using IDT xGen Lockdown probes). However, the clever reuse of oligo synthesis pools I have not seen before and is a highly valuable contribution allowing this approach to become feasible to labs where the

same enrichment is required many times (e.g., > 100). The figures are beautifully presented and it was refreshing to see all data and code available in public repositories. I believe this work will be of great interest to the growing long-read sequencing community and an excellent fit for Nature Communications.

I had the following comments that I would like addressed:

Line 106: Where does the >10,000 reactions come from? I understand the mixed oligo pool can be used numerous times, but I cannot see how there would be enough initial material in the non-biotinylated pool. Some elaboration on this point would be helpful and if not tested, perhaps using the term “in theory” would be more appropriate.

Line 111: Why is PCR amplification required? Wouldn't this add more bias to the quantification step? I also find it surprising that such a high correlation is seen after the level of amplification performed. Were you not surprised? And why do you think this is?

Line 133: When comparing the fold-enrichment against the whole-transcriptome sequencing, why was quantification not assessed on this data set. You later look at using the ERCC standard, but those are synthetic and not particularly long and you have the data from a more realistic setting to present? Is there a reason for this, and how well does the quantification of your enriched sample compare to the whole transcriptome dataset? (I know you later go on to look at this with other data sets, but I was confused as to why this initial one was not assessed as well?)

Line 320: It is fairer to say that while the setup costs are comparable to existing approaches, reagents for a specific enrichment panel can be used many times, reducing the overall costs. The costs you provide per reaction are with what I feel is a very high reuse, which in many cases (e.g., where different problems are studied) may not be true.

General: A major advantage of emerging RNA sequencing approaches with long-read approaches like nanopore is the ability to directly read the modifications that are present. This information is lost when using this method. It would be useful to perhaps highlight this limitation in the discussion to ensure this limitation is understood.

Title: Given that long-read RNA sequencing is often confused with Direct RNA sequencing, it would be more appropriate for the title to be: “TEQUILA-seq: A versatile and low-cost method for targeted long-read cDNA sequencing”.

Response to Reviews

We thank the reviewers for their highly positive comments about our manuscript. All three reviewers were enthusiastic about our work. We have carried out new experiments and analyses and revised our manuscript accordingly to address the reviewers' questions and suggestions. Most importantly, in response to a reviewer's question, we have implemented sample barcoding and multiplexing, which further increase the scalability and decrease the cost of TEQUILA-seq. Overall, we are pleased that these revisions have further improved the quality and impact of this manuscript.

Reviewer #1:

*In their paper "TEQUILA-seq: A versatile and low-cost method for targeted long-read RNA sequencing", Wang et al. introduce the a method for generating low cost enrichment probes using nicking-
endonuclease (nickase)-triggered isothermal strand displacement amplification (SDA). They benchmark the use of these probes by enriching full-length cDNA samples.*

First, the authors compare TEQUILA probes to IDT xGen probes on a panel of 10 genes which they enriched from a commercially available sample of human brain RNA. Next they test whether enrichment affects relative quantification using spike-in controls. Further they enriched a panel of cancer genes from cDNA of 40 breast cancer cell lines and performed isoform-level analysis on the resulting data using their new "Espresso" analysis pipeline.

I think the easy generation of cheap enrichment probes as described by the manuscript is a great addition to the field. This is something I would personally try and use and I believe there exists a real use case for this. The fact that these probes seem to perform almost identically to xGen probes is remarkable.

Using full-length cDNA analysis for the benchmarking of these probes is interesting because that represents a still somewhat niche application of target enrichment. Also, analyzing full-length cDNA comes with the added challenge of isoform identification and quantification which can make it difficult to disentangle the performance of the enrichment probes and the cDNA sequencing and analysis approach.

There are some really cool things in the manuscript, like explaining alternative isoform expression with structural variants in cell lines and overall, I think this is a strong manuscript.

I do have some points that I would like to see addressed or considered:

We thank the reviewer for the highly positive comments about our work. The reviewer commented that "the easy generation of cheap enrichment probes as described by the manuscript is a great addition to the field", and that "this is something I would personally try and use and I believe there exists a real use case for this". The reviewer raised several points to be addressed or considered. We have revised our manuscript to address the reviewer's questions and suggestions, as detailed below.

1) Because there is a considerable focus on the cost-effectiveness of the probes, I would like to see more detail on the process of probe generation:

A) There is some information on yields in Supplemental Figure 1 but I would like to see actual data on the yields of probe generation reactions. What is the range of material individual reactions produce? I think having that information in a table or figure would be very useful.

We have performed a series of probe synthesis reactions for the 468 actionable cancer genes, using varying amounts of template oligo pool ranging from 2 ng to 10 ng, and varying incubation times ranging from 1 hr to 16 hr. For each of these combinations, TEQUILA probes were synthesized using fixed

concentrations of enzymes (0.8 U/ μ l of Klenow Fragment (3'-5' exo-) DNA polymerase, 3 nM (0.04 U/ μ l) of Nt.BspQI), 0.6 mM dNTPs, and other reaction components as described in the Methods section. We quantified the resulting probe yield using Nanodrop (Fig. R1). With increasing amounts of template oligo pool, we observed a corresponding increase in probe yield. Moreover, the probe yield increased with longer incubation times, although the rate of increase slowed down after 4 hours. This may be explained by the fact that the enzymes and dNTPs used in the probe synthesis reaction became exhausted over time.

Figure R1: Probe yield of individual synthesis reactions measured by Nanodrop. a and b show probe yield from different amounts of template oligo pool (ranging from 2 ng to 10 ng) and incubation times (ranging from 1 hr to 16 hr).

Importantly, using an input of only 2 ng of template oligo pool, we were able to generate up to 25 μ g of TEQUILA probes, which can be used for 250 capture reactions, with 100 ng of probes per capture reaction. For a panel of 6,000 probes, the total amount of non-biotinylated template oligo pool purchased from commercial vendors typically ranges from 50 ng to more than 200 ng based on our experience, depending on batch variations during array-based oligo synthesis. Assuming that 2 ng of template oligo pool is regularly used per probe synthesis reaction, the total amount of template oligo pool can be used for 25 to 100 TEQUILA probe synthesis reactions. Therefore, the total amount of TEQUILA probes can potentially be used for 6,250 (250×25) to 25,000 (250×100) capture reactions. Overall, the results demonstrate a high probe synthesis yield. We have included these new results and Fig. R1 as a new supplementary figure in the revised manuscript.

B) Size of the generated probes: Do the sizes of the generated probes match the length of the template oligos? Gel or tapestation images would be great for this.

We followed the reviewer's suggestion and investigated the migration pattern of TEQUILA probes using a TapeStation assay (Fig. R2). Specifically, we analyzed different sets of probes and oligos for the 10 human brain genes, including xGen-lockdown probes (IDT, 120 nt, mono-biotinylated), template oligo pool (Twist Bioscience, 150 nt, non-biotinylated), and TEQUILA probes (expected size: 120 nt, multi-biotinylated). As expected, single-stranded DNA (xGen-lockdown probes, Lane 1; template oligo pool, Lane 2) migrated differently compared to double-stranded DNA (DNA ladders, lane M). Moreover, TEQUILA probes (Lane 3) had a wider range of migration compared to mono-biotinylated xGen-lockdown probes (Lane 1) and non-biotinylated template oligo pool (Lane 2), consistent with the fact that

TEQUILA probes are multi-biotinylated and that DNA migration speed decreases as biotinylation level increases (PMID: 1288639).

To eliminate biotinylation level as a confounding factor in assessing the size of TEQUILA probes using the TapeStation assay, we generated non-biotinylated TEQUILA probes by replacing biotin-dUTPs with dTTPs in the TEQUILA probe synthesis reaction. The migration of non-biotinylated TEQUILA probes (Lane 4) was concentrated in a much narrower range compared to biotinylated TEQUILA probes (Lane 3). As expected, non-biotinylated TEQUILA probes (Lane 4) migrated faster than mono-biotinylated xGen-lockdown probes (Lane 1) and non-biotinylated template oligo pool (Lane 2). We also observed that non-biotinylated TEQUILA probes (Lane 4) had a moderately wider migration range compared to the template oligo pool (Lane 2). This was expected, because the Klenow Fragment (3'-5' exo-) DNA polymerase used for TEQUILA probe synthesis has a strand displacement activity and may generate synthesis byproducts that result in variations in oligo size. Nonetheless, despite these potential synthesis byproducts, TEQUILA probes have a high capture performance as demonstrated by extensive benchmark analyses conducted in this work.

As an additional control, we incubated biotinylated TEQUILA probes with streptavidin before loading. All TEQUILA probes (Lane 5) were trapped in the loading well, indicating high biotinylation levels of TEQUILA probes.

Figure R2: High-sensitivity D1000 TapeStation image showing migration patterns of probes and oligos. Lane M: DNA ladders; Lane 1: xGen-Lockdown probes (IDT, 120 nt, mono-biotinylated); Lane 2: template oligo pool (Twist Bioscience, 150 nt, non-biotinylated); Lane 3: TEQUILA probes (expected size: 120 nt, multi-biotinylated); Lane 4: non-biotinylated TEQUILA probes (expected size: 120 nt); Lane 5: TEQUILA probes incubated with streptavidin prior to loading.

2) *Enrichment reactions usually use some form of blocking oligos. However the manuscript does not mention those. Were the enrichment reactions indeed performed without blocking oligos?*

We did not use blocking oligos in any of the experiments presented in the original manuscript. We agree with the reviewer that blocking oligos are commonly used in enrichment reactions (PMID: 24705597, 32393825). To investigate whether TEQUILA-seq could benefit from the use of blocking oligos, we designed three types of blocking oligos, including oligos targeting the adapters used in probe synthesis (1 nmol of CGAAGAGCCCTATAGTGAGTCGTATTAG/3SpC3/), oligos targeting the primers for cDNA

amplification (1 nmol of AAGCAGTGGTATCAACGCAGAGT), and oligo(dT)18 with a three-carbon spacer at the 3' end targeting the poly(A) tail (1 nmol of TTTTTTTTTTTTTTTTTT/3SpC3/).

Next, we performed TEQUILA-seq of 468 actionable cancer genes on 4 breast cancer cell lines (HCC1806, MDA-MB-157, AU-565, and MCF7) with or without using blocking oligos. It is worth noting that as we have become more experienced in performing TEQUILA-seq over time, there was a modest increase in the on-target rates even without blocking oligos in these new experiments, as compared to the on-target rates reported in the original manuscript. Nonetheless, by comparing the results of TEQUILA-seq with or without using blocking oligos, we found that the use of blocking oligos further improved the on-target rates by 2.8% to 5.6% (Fig. R3). Overall, our results demonstrate that using blocking oligos can modestly improve the capture performance of TEQUILA-seq. We have included these new results and Fig. R3 as a new supplementary figure in the revised manuscript.

Figure R3: Target enrichment of 468 actionable cancer genes in 4 breast cancer cell lines using TEQUILA-seq with or without blocking oligos. For each cell line, TEQUILA-seq (with or without blocking oligos) and whole-transcriptome nanopore 1D cDNA sequencing (non-capture control) libraries were prepared from the same biological replicate. Each bar shows the percentage of reads mapped to the 468 genes.

3) Does enrichment affect read lengths? The methods section show different SPRI bead ratios are used after initial cDNA synthesis (0.8) and after PCR following enrichment (0.7). What is the reason for and result of this?

We used different volume ratios of SPRIselect beads (0.8 after initial cDNA synthesis and 0.7 after PCR following enrichment) based on the optimized parameters of each individual step according to our experience. Although the SPRI bead ratios used in these two steps were slightly different, we used the same 0.7 ratio of SPRIselect beads in the subsequent step of cDNA purification (after end repair/dA-tailing) for both whole-transcriptome 1D cDNA sequencing and TEQUILA-seq. This cDNA purification step affects the final library size, so there is no difference in size selection for whole-transcriptome 1D

cDNA sequencing and TEQUILA-seq due to the choice of SPRI bead ratios. We appreciate the reviewer for bringing up this issue and have additionally revised the Methods section to make a correction about the bead ratio used for cDNA purification (revised text highlighted in bold below).

*“Briefly, cDNA products were end-repaired and dA-tailed using NEBNext Ultra II End Repair/dA-Tailing Module (NEB, # E7546) by incubating at 20°C for 20 min and 65°C for 20 min. The cDNA was then purified with **0.7× volumes of SPRIselect beads** and eluted in 60 µl of nuclease-free water.”*

To investigate whether target enrichment affects read lengths, we compared the distributions of read lengths between whole-transcriptome 1D cDNA sequencing libraries and TEQUILA-seq libraries of the 468 actionable cancer genes across the four breast cancer cell lines (HCC1806, MDA-MB-157, AU-565, and MCF7) (Fig. R4). As expected, the reads mapped to target genes were generally longer than the reads mapped to all human genes (Fig. R4, panel a vs. panel b), consistent with the median transcript length of target genes compared to that of all human genes (1,100 bp vs 876 bp, based on GENCODE annotation v34lift37). Moreover, the read length distributions of whole-transcriptome 1D cDNA sequencing libraries and TEQUILA-seq libraries were largely comparable for reads mapped to target genes (Fig. R4, panel b), indicating that target enrichment did not substantially affect read lengths.

Figure R4: Read length distributions of whole-transcriptome 1D cDNA sequencing libraries and TEQUILA-seq libraries of 468 actionable cancer genes across 4 breast cancer cell lines.

4) Often many samples can be multiplexed during target enrichment to minimize costs but also batch effects of enrichment and sequencing. Why weren't samples in this study indexed and pooled before enrichment? Even R9.4/guppy4 should be accurate enough to index full-length cDNA with for example 20nt sample barcodes in oligodT primers. In the unlikely case that pooling samples before enrichment using TEQUILA probes isn't possible, this should be mentioned when cost per sample is discussed.

This is a great question. In the work reported in the original manuscript, we sequentially sequenced two replicates of TEQUILA-seq libraries for each breast cancer cell line on one MinION flow cell, using ONT's flow cell wash kit following ONT's recommendation. Specifically, we used the flow cell wash kit to wash out the first library and then refresh the flow cell for the second library to be loaded and sequenced. We have added the following text to the Methods section of the revised manuscript to clarify this issue:

“For TEQUILA-seq of 468 actionable cancer genes across 40 breast cancer cell lines, the two replicates of TEQUILA-seq libraries for each breast cancer cell line were sequentially sequenced on one MinION flow cell. The flow cell wash kit (EXP-WSH004) was used to wash out the first library and then refresh the flow cell for the second library to be loaded and sequenced. All fluids from the waste channel were removed through waste port 1. Next, a mix of 2 µl of wash mix (WMX, containing DNase I) and 398 µl of wash diluent (DIL) was loaded into the flow cell priming port and incubated for 90 min. After digestion, all fluids from the waste channel were removed through waste port 1. A second library was then loaded for sequencing by following the instructions in the “Priming and loading the flow cell” section of the 1D ligation kit protocol (SQK-LSK109).”

We fully agree with the reviewer that barcoding and multiplexing samples for TEQUILA-seq would be an important approach to further reduce costs and minimize batch effects of enrichment and sequencing. To implement the multiplexing capability of TEQUILA-seq, we utilized the newly upgraded ONT native barcoding kit (NBD114) for the newly released R10.4 version of ONT sequencing chemistry. We added sample barcodes and performed multiplexed TEQUILA-seq on 4 breast cancer cell lines (HCC1806, MDA-MB-157, AU-565, and MCF7) on an R10.4 MinION flow cell. We generated 7.2 million reads. 88.4% of reads can be uniquely assigned to one of the 4 cell lines with correct sample barcodes, indicating excellent performance of demultiplexing. Compared to TEQUILA-seq of the 4 breast cancer cell lines without barcoding and multiplexing, TEQUILA-seq with barcoding and multiplexing resulted in similar on-target rates (Fig. R5, panel a). Moreover, TEQUILA-seq samples with or without multiplexing for each breast cancer cell line were clustered together based on transcript expression levels of 468 actionable cancer genes (Fig. R5, panel b).

Figure R5: Barcoding and multiplexing TEQUILA-seq libraries using the nanopore native barcoding kit. a. Percentage of reads mapped to 468 actionable cancer genes in 4 breast cancer cell lines using TEQUILA-seq (with or without multiplexing) and whole-transcriptome nanopore 1D cDNA sequencing (non-capture control). b. Principal component analysis using estimated abundances of all transcript isoforms across 468 genes in 4 cell lines. Each data point represents one TEQUILA-seq sample for a cell line.

We are very pleased by these new results. The capability for sample barcoding and multiplexing represents a significant new advance, as it further increases the scalability and decreases the cost of TEQUILA-seq. It also allows TEQUILA-seq to harness the higher throughput of the ONT PromethION platform. We have included these new results and Fig. R5 as a new supplementary figure in the revised manuscript.

5) cDNA sequencing on the MinION is notorious for batch effects. Read numbers can vary greatly (and do in this study according to Table S15) and read length distributions can be all over the place. This can affect isoform analysis. Further, it would affect the quantification of transcripts. This leads me to a two questions:

What are the read length distributions for all experiments in this manuscript? Violin plots would be nice but a table with Median and 25th and 75th percentile would do.

If read lengths are different between experiments I would like to see a discussion how this is going to affect isoform analysis.

Following the reviewer's suggestion, we investigated the read length distributions for all datasets presented in our manuscript. Overall, the read length distributions were consistent and comparable between different experiments for the same TEQUILA-seq gene panel. For example, the read length distributions of TEQUILA-seq of 468 actionable cancer genes were comparable across 40 breast cancer cell lines (Fig. R6).

Figure R6: Read length distributions of TEQUILA-seq of 468 actionable cancer genes across 40 breast cancer cell lines. The middle band of the box indicates the median, and the top and bottom of the box indicate the third and first quartiles, respectively.

6) The quantification of the 5 enriched long SIRVs is interesting for several reasons. First, what 5 SIRVs were selected and what is their respective length. How does this compare to the read length of the entire sequencing run (before and after enrichment) and more specifically, how does this relate to the lengths of

the reads aligning to each SIRV? If the length of SIRVs and its aligning reads is not in agreement, how exactly is quantification performed by ESPRESSO?

Long SIRVs are an interesting example because they all have entirely unique sequences meaning transcript length could be irrelevant to their quantification (i.e. sequencing only the most 3' 50nt of them would be enough to identify them - an edgecase quite different from comparing isoforms within a gene).

We selected five long SIRVs for benchmarking TEQUILA-seq, including SIRV4003 (4,031 nt), SIRV6003 (6,030 nt), SIRV8003 (8,030 nt), SIRV10002 (10,031 nt), and SIRV12003 (12,029 nt). Our intention behind using long SIRVs was to investigate if there was a preferential enrichment of long transcripts because they had more probes targeting them (e.g., more probes targeting a 12 kb transcript than a 4 kb transcript). Based on results presented in Fig. 2b, we did not observe such transcript-length dependent bias for target enrichment.

As expected, our analysis revealed that reads mapped to the long SIRVs were generally longer than reads mapped to other target genes, such as the 468 actionable cancer genes. However, the read lengths for the long SIRVs may be affected by technical issues such as the elongation ability of reverse transcriptase and DNA polymerase, as well as the nanopore sequencing process. In fact, the majority of reads mapped to the five long SIRVs were not full length, and this observation was made in whole-transcriptome 1D cDNA sequencing without any target enrichment (Fig R7, top panel). Nonetheless, compared to whole-transcriptome 1D cDNA sequencing, TEQUILA-seq preserved the read length distribution of the five long SIRVs (Fig R7, bottom panel) and enriched each transcript to a comparable degree (Fig. 2b).

We should also note that ESPRESSO employs an expectation-maximization (EM) algorithm for transcript quantification incorporating both full-length and non-full-length reads. The EM algorithm is a standard algorithm for transcript quantification using sequence fragments, initially developed in 2006 by one of our senior authors Dr. Xing on expressed sequence tags (ESTs) (PMID: 16757580) and later widely adopted for short-read and long-read RNA-seq transcript quantification. Details of ESPRESSO along with comprehensive benchmark analyses of this tool were presented in our recent publication (PMID: 36662851).

Figure R7: Length distributions of reads mapped to target long SIRVs in whole-transcriptome 1D cDNA sequencing libraries and TEQUILA-seq libraries. The dashed vertical line indicates the transcript length

of each target long SIRV. Note that the read counts shown for these two types of libraries have different scales.

Reviewer #2:

Long-read sequencing platforms, such as Pacific Biosciences and Oxford Nanopore Technologies are powerful and enabling technologies that allows for full-length transcript sequencing. Despite their potentials in a variety of biomedical and clinical applications, one major bottleneck of the long-read sequencing platforms is that their throughput is several orders of magnitude lower than that of short-read platforms. One way to overcome this bottleneck is through targeted sequencing for predefined gene panels. However, existing solutions for targeted long-read RNA-seq are either expensive, or difficult to implement. In the current study, Wang et al. describes the development of TEQUILA-seq, a versatile, easy-to-implement, and low-cost method for targeted long-read RNA-seq. The major innovation of TEQUILA-seq is that it uses nicking-endonuclease- triggered isothermal strand displacement amplification to synthesize large quantities of biotinylated capture oligos from an array-synthesized pool of non-biotinylated oligo templates. As a result, TEQUILA-seq reduces the per-reaction cost of targeted capture by 2-3 orders of magnitude, in comparison with the standard commercial solution. They further applied TEQUILA-seq to a panel of 468 actionable cancer genes across representative breast cancer cell lines and identified transcript isoforms enriched in specific subtypes and discovered novel transcript isoforms in extensively studied cancer genes. TEQUILA-seq is a long-awaited method that will greatly facilitate the wide adoption of targeted long-read RNA-seq in different basic research and clinical application settings.

My detailed comments are listed as follows.

We thank the reviewer for the highly positive comments about our work. The reviewer commented that “TEQUILA-seq is a long-awaited method that will greatly facilitate the wide adoption of targeted long-read RNA-seq in different basic research and clinical application settings”. We have revised our manuscript to address the reviewer’s questions and suggestions, as detailed below.

1. The key innovation of TEQUILA-seq is a new low-cost method for generating large quantities of biotinylated capture oligos. This innovation will potentially significantly reduce the cost of targeted DNA-seq as well. The authors only briefly mentioned the adoption of this method for targeted DNA-seq in the discussion part, which is not very visible. It would be worth expanding on the potential impact of this new biotinylated capture oligo generation method in both introduction and discussion section, on targeted long/short-read DNA-seq for population-scale genetics study and precision medicine.

We appreciate the reviewer’s suggestion. We would like to note that our manuscript is primarily focused on the development, benchmarking, and application of TEQUILA-seq for targeted long-read RNA-seq – an application of broad interest and utility. While TEQUILA probes can also be used for targeted DNA sequencing, as we noted in the Discussion section of the original manuscript, we are concerned that an expanded discussion of this application in the Introduction section could dilute our message and make the manuscript too diffuse. Nonetheless, in light of the reviewer’s suggestion, we have added the following text to the end of the Introduction section, to briefly describe the applications of TEQUILA probes beyond targeted long-read RNA-seq:

“Moreover, TEQUILA probes are compatible for both targeted RNA and DNA sequencing, on both long-read and short-read sequencing platforms. The ability to easily generate large quantities of biotinylated capture oligos for any target panel at a low cost and a high efficiency can facilitate large-scale and population-level studies for a wide range of basic, translational, and clinical applications.”

2. The ERCC standard synthetic transcripts of unique sequences and diverse concentrations were used to evaluate TEQUILA-seq's capability of preserving and quantification of target transcripts. The use of synthetic transcript pool alone for evaluation has limitations because it does not reflect the complexity of the human transcriptome in cells/tissues. Therefore, it is important to evaluate TEQUILA-seq's performance of preserving and quantification of target transcripts by using the synthetic transcripts as the spike-in that is mixed with the RNAs extracted human cells/tissues.

We apologize for any confusion caused by our original text. What the reviewer suggested was exactly what we did for the results presented in Fig. 2. We used synthetic transcripts (ERCCs and long SIRVs) as spike-ins mixed with RNAs extracted from the human SH-SY5Y neuroblastoma cell line. To clarify this issue, we have updated the following text in the Results section of the revised manuscript (revised text highlighted in bold below):

“To evaluate the performance of TEQUILA-seq for transcript detection and quantification, we used RNAs of human SH-SY5Y neuroblastoma cells mixed with synthetic transcripts as spike-ins, including External RNA Controls Consortium (ERCC) standards and Spike-In RNA Variants (SIRVs). TEQUILA-seq was performed using three sets of probes targeting selected ERCC and SIRV synthetic transcripts as well as 221 human genes encoding splicing factors (see below). First, we assessed the extent to which TEQUILA-seq improves detection of transcript isoforms of target genes by using the ERCC standards.”

3. The SIRVs comprising 15 synthetic transcripts of equimolar concentrations that cover transcript lengths from 4,000 to 12,000 nt was used to determine whether TEQUILA-seq data exhibit any transcript length-dependent biases. To justify the evaluation using SIRVs, further information needs to be provided. For example, what is the length distribution of isoforms for typical human genes and how representative the length distribution of SIRVs isoforms is among human genes? How far we can extrapolate the results from SIRV to other human genes? It is possible that the option of synthetic isoforms for this type of evaluation is limited. It is thus important to discuss about the limitation of using SIRVs for evaluation.

We appreciate the reviewer's concern. We agree with the reviewer that the long SIRVs, or any set of target transcripts (synthetic or real), have their limitations. This was exactly why in this work we used five distinct target panels (10 brain genes, ERCCs, long SIRVs, 221 genes encoding splicing factors, and 468 actionable cancer genes) to comprehensively benchmark the performance of TEQUILA-seq. We selected five long SIRVs for benchmarking TEQUILA-seq, including SIRV4003 (4,031 nt), SIRV6003 (6,030 nt), SIRV8003 (8,030 nt), SIRV10002 (10,031 nt), and SIRV12003 (12,029 nt). These long SIRVs were meant to serve a *single* purpose – our goal was to investigate if there was a preferential enrichment of long transcripts because they had more probes targeting them (e.g., more probes targeting a 12 kb transcript than a 4 kb transcript). Based on results presented in Fig. 2b, we did not observe such transcript-length dependent bias for target enrichment.

Besides long SIRVs, we also used four other target panels to benchmark TEQUILA-seq for transcript detection and quantification (10 brain genes, ERCCs, 221 genes encoding splicing factors, and 468 actionable cancer genes). These target panels included both synthetic and real transcripts, and covered diverse distributions of transcript lengths and concentrations. For example, the ERCC transcripts were 281 to 2,036 nt in length, largely overlapping with the length distribution of human protein-coding transcripts (PMID: 31164174). To clarify this, we added the following text to the Results section of the revised manuscript:

“These ERCC transcripts were 281 to 2,036 nt in length, largely overlapping with the length distribution of human protein-coding transcripts (PMID: 31164174).”

4. In the overview of TEQUILA-seq, it says “full-length cDNAs are synthesized from poly(A)+ RNAs by reverse transcription and PCR amplification”. As there are many non-polyA transcripts in the human transcriptome, it would be important to describe the situation for non-poly (A)+ RNAs, even if the current study focuses on poly(A)+ ones.

We have added the following text to the Discussion section of the revised manuscript:

“While the current TEQUILA-seq protocol based on the standard nanopore 1D cDNA sequencing workflow is designed for poly(A)+ RNAs, it can be easily adapted for non-poly(A)+ RNAs with minor modifications, such as poly(A)+ tailing of non-poly(A)+ RNAs.”

5. By using TEQUILA-seq, the authors identified transcript isoforms enriched in specific subtypes and discovered novel transcript isoforms in extensively studied cancer genes in breast cancer cell lines. As the cell line could be from tumors, how many of these discoveries have supporting evidence from RNA-seq data generated in human tumors, e.g., from TCGA breast cancer data?

The primary goal of our manuscript is to report TEQUILA-seq as a versatile and low-cost method for targeted long-read RNA-seq. While the comparison of TEQUILA-seq data on breast cancer cell lines with TCGA RNA-seq data on breast cancer tissues is potentially of interest, we feel it is peripheral to our study and beyond the scope of our manuscript. There are also multiple biological and technical reasons that make such a comparison less meaningful. TEQUILA-seq data were generated on cancer cell lines, while the TCGA RNA-seq data were generated on bulk tumor tissues with a mixed cell type composition. TEQUILA-seq data were generated by long-read RNA-seq, while the TCGA RNA-seq data were generated by short-read RNA-seq which has an inherent limitation for transcript isoform analysis. Additionally, the “tumor aberrant” transcript isoforms found by TEQUILA-seq in individual breast cancer cell lines largely arose from “personalized” small or large variants in DNA (e.g., as shown in *TP53*, *NOTCH1*, and *RBI*). For these transcript isoforms, unless a particular TCGA tumor tissue carries the same variant above a certain clonality threshold, we do not expect to detect them in the TCGA RNA-seq data.

Nonetheless, we sought to examine the transcript isoforms of three genes (*DNMT3B*, *FGFR2*, and *SESNI*) in the TCGA breast cancer (BRCA) RNA-seq data. These three genes had breast cancer subtype-associated transcript isoforms, based on our TEQUILA-seq analysis of 40 breast cancer cell lines. We used STAR (v2.6.1d) and Cufflinks (v2.2.1) to quantify transcript expression levels of these three genes based on TCGA short-read RNA-seq data. For *DNMT3B* and *SESNI*, the transcript isoforms identified by TEQUILA-seq as enriched in the basal B subtype of breast cancer cell lines were also found to have higher isoform proportions in TCGA breast cancer tissues that were classified as “basal”, indicating concordant results between the two datasets (Fig. R8, panel a and b). For *FGFR2*, the transcript isoform identified by TEQUILA-seq as enriched in the basal B subtype had almost no detectable transcript expression in the TCGA RNA-seq data across the subtypes, with the highest median transcript expression level (measured in FPKM) being 0.11 in the HER2 enriched subtype (Fig. R8, panel c). We should note that the transcript isoform switch of *FGFR2* detected by TEQUILA-seq across breast cancer cell lines is a hallmark of epithelial-to-mesenchymal transition associated with the invasiveness and aggressiveness of breast cancer cells (PMID: 19285943). The failure to detect this cell type-specific transcript isoform switch in TCGA RNA-seq data largely reflects the limitation of transcriptome analysis using bulk tumor tissues with a mixed cell type composition, in which the vast majority of cells are likely epithelial.

Figure R8: Isoform abundances and proportions of TEQUILA-seq identified subtype-associated transcript isoforms in *DNMT3B*, *SESN1*, and *FGFR2*, in TCGA breast cancer RNA-seq data. The middle band of the box indicates the median, and the top and bottom of the box indicate the third and first quartiles, respectively.

6. In the method section, it is not very clear how the probe design was performed. What methods/software were used?

To address this question, we have added the following text to the Methods section of the revised manuscript:

“The remaining 120 nt were designed to tile across all annotated exons, including UTRs, of target genes with 1× tiling density, by using the xGen Hyb Panel Design Tool (<https://www.idtdna.com/pages/tools/xgen-hyb-panel-design-tool>). Specifically, BED/FASTA files, gene symbols, or transcript accession numbers of target genes were uploaded as input, and "Homo sapiens (Human) NCBI GRCh37.p13 (hg19)" was used as the reference. Probe length and probe tiling density were set to 120 nt and 1x, respectively. A spreadsheet of 120 nt oligo sequences was generated as output along with information on target base counts, probe counts, and coverage percentage.”

7. It says “We defined tumor aberrant transcript isoforms as transcript isoforms that are present at significantly elevated proportions in at least one but no more than 4 (i.e. ≤10%) breast cancer cell lines”. It is not clear why these isoforms are defined as “tumor aberrant transcript” because there was no comparison between tumor cells and normal cells here and these transcripts could show high expression in normal cells/tissues as well. Because these aberrantly expressed isoforms only occurs in <10% breast cancer cell lines, cautions need to be taken about whether they are true aberrantly expressed isoforms or they were identified due to some technical artifacts from sequencing data. Are these transcripts associated/potentially caused by some specific mutations in the small proportion of cell lines?

We appreciate the reviewer’s comments. We would like to clarify that our definition of “tumor aberrant transcript isoforms” is essentially the same as “outlier transcript isoforms”, as they were required to be

present at significantly elevated proportions in a subset of breast cancer cell lines (in our study, at least one but no more than 4 (i.e. $\leq 10\%$)). We sought to define and analyze such tumor aberrant transcript isoforms through an outlier analysis, as opposed to through a cancer vs normal cell line comparison, primarily because the latter was much less interpretable due to the lack of true “normal” cell lines for such a comparison. To address the concern that these identified tumor aberrant transcript isoforms may arise from technical artifacts, we had taken multiple measures to ensure the reliability of our results. Importantly, we required that the identified transcript isoforms must be present as outliers in *both replicates* of a given cell line, and we applied stringent criteria to filter out potential false positives (as detailed in the Methods section).

The reviewer’s intuition that these tumor aberrant transcript isoforms are likely caused by specific mutations in a small proportion of cancer cell lines is spot on. In fact, the initial motivation for our analysis of tumor aberrant transcript isoforms was to identify transcript isoforms arising from “personalized” variants in tumor DNA. This was exactly what we found in our results, as shown for multiple genes in which the identified tumor aberrant transcript isoforms could be attributed to small or large variants in DNA (e.g. *TP53*, *NOTCH1*, and *RBI*). The fact that these tumor aberrant transcript isoforms could be explained by DNA variants provides further evidence that they were real and not due to technical artifacts. To clarify our definition of tumor aberrant transcript isoforms, we have added the following text to the Results section of the revised manuscript:

“We note that the tumor aberrant transcript isoforms identified based on this definition were essentially “outlier” transcript isoforms with elevated proportions in a small subset of breast cancer cell lines. To eliminate potential technical artifacts and ensure the reliability of our results, we required that the identified tumor aberrant transcript isoforms must be present as outliers in both replicates of a given cell line, and we applied stringent criteria to filter out potential false positives (Methods).”

8. *Is there any literature knowledge about the biological functions of the identified subtype specific isoforms?*

For *DNMT3B*, the transcript isoform identified by TEQUILA-seq as enriched in the basal B subtype involves skipping of exons 21 and 22 that disrupts the C-terminal catalytic domain; thus, the encoded protein isoform is known to be enzymatically inactive. We discussed the functional consequence and cited the relevant literature in the original manuscript. In response to the reviewer’s question, we have added the following text to the Results section of the revised manuscript, to discuss the literature knowledge about the subtype-associated transcript isoforms in *FGFR2* and *SESNI*:

“The transcript isoform switch of FGFR2 detected by TEQUILA-seq across breast cancer subtypes is a hallmark of epithelial-to-mesenchymal transition associated with the invasiveness and aggressiveness of breast cancer cells (PMID: 19285943). This transcript isoform switch involves two mutually exclusive exons that encode different versions of the ligand binding domain (Supplementary Fig. 4), generating FGFR2 protein isoforms with distinct ligand binding specificities (PMID: 16597617). The transcript isoform switch of SESNI involves mutually exclusive uses of alternative first exons (Supplementary Fig. 5) and has been associated with isoform-specific transcript induction by genotoxic stress in a p53-dependent manner (PMID: 9926927).”

9. *It was found that many tumor aberrant transcript isoforms identified by TEQUILA-seq contain premature termination codons, which would target the transcript isoform for degradation via nonsense-mediated (NMD) mRNA decay. It remains unclear why the NMD-targeted isoforms show higher expression than other isoforms that are not NMD-targeted. Do these genes with dominant NMD-targeted isoforms show lower overall expression than the same genes with less NMD-targeted isoforms in other cell lines? It would be important to clarify and discuss the related issues.*

We appreciate the reviewer's question. Nonsense-mediated mRNA decay (NMD) degrades transcripts containing premature termination codons (PTCs). However, the efficiency of NMD varies considerably across transcripts and tissue types depending on various factors such as PTC position (PMID: 27618451), concentration of NMD factors (PMID: 34893608), tissue and cell type (PMID: 25954003, 17625509), and physiological condition (PMID: 26397022). As a result, PTC-containing transcript isoforms can frequently escape NMD and be readily detected in transcriptome data (PMID: 25954003, 27618451). As the isoform proportions of NMD-targeted transcript isoforms identified by TEQUILA-seq represent their steady-state profiles combining the effects of transcript production and degradation, it is not surprising that certain NMD-targeted transcript isoforms have considerable isoform proportions, sometimes even higher than other non-NMD-targeted transcript isoforms of the same genes. Interestingly, and consistent with the reviewer's intuition, the cell line with multiple NMD-targeted *TP53* transcript isoforms (HCC1599) was among the cell lines with the lowest steady-state *TP53* gene expression levels (Fig. 4c). To further confirm that these *TP53* transcript isoforms were NMD-targeted, we performed a new experiment by treating HCC1599 cells with NMD inhibitor SMG1i (compound 11j). We observed a substantial increase in the isoform abundances of the two PTC-containing transcript isoforms, especially the predominant transcript isoform containing the retained intron (Fig. R9, top band). These data indicate that these transcript isoforms were indeed targeted and degraded by NMD at the steady state in HCC1599 cells.

Figure R9: Elevated transcript expression levels of two PTC-containing transcript isoforms of *TP53* upon NMD inhibition in HCC1599 cells, as indicated by RT-PCR analysis. The 689-bp product corresponds to the predominant novel transcript isoform containing a retained intron, while the 170-bp product corresponds to a second, minor novel transcript isoform using a novel 3' splice site. *GAPDH* was used as the loading control for RT-PCR. Western blot analysis of phosphorylated UPF1 (pUPF1) was used to measure NMD activity, and β -actin was used as the loading control for Western blot.

In response to the reviewer's question, we have added the following text to the Results section of the revised manuscript:

“We should note that the efficiency of NMD varies considerably across transcripts and tissue types depending on various factors such as PTC position (PMID: 27618451), concentration of NMD factors (PMID: 34893608), tissue and cell type (PMID: 25954003, 17625509), and physiological condition (PMID: 26397022). As a result, PTC-containing transcript isoforms can frequently escape NMD and be readily detected in transcriptome data (PMID: 25954003, 27618451). Interestingly, the cell line with multiple NMD-targeted TP53 transcript isoforms (HCC1599) was among those with the lowest steady-state TP53 gene expression levels across the 40 breast cancer cell lines (Fig. 4c), consistent with the expected effect of NMD on steady-state gene expression level. To further confirm that these TP53 transcript isoforms were NMD-targeted, we treated HCC1599 cells with NMD inhibitor SMGIi (compound IIj). We observed a substantial increase in the isoform abundances of the two PTC-containing transcript isoforms, especially the predominant novel transcript isoform containing the retained intron (Supplementary Fig. 10). These data indicate that these transcript isoforms were indeed targeted and degraded by NMD at the steady state in HCC1599 cells.”

10. The current study relies on the software ESPRESSO. Has it been peer-reviewed?

Yes, the ESPRESSO software for robust discovery and quantification of transcript isoforms using long-read RNA-seq data was reported in a peer-reviewed publication in Science Advances in January 2023 (PMID: 36662851). We have updated the citation for ESPRESSO in the revised manuscript.

Reviewer #3:

In this paper a new methodology called TEQUILA-seq is presented which aims to reduce the cost of targeted RNA enrichment through the reuse of oligo pools. The authors comprehensively demonstrate that their methodology is able to greatly enrich required targets and that quantification information about those targets is not lost during the process. This is somewhat surprising given the bias that is sometimes seen in some of these studies. That aside, the experiments they perform include a mixture of real and synthetic samples and the data presented backs up their claims. Furthermore, they are able to apply the approach to 40 breast cancer cells lines and efficiently identify interesting transcript iso-forms enriched in key subtypes.

Overall, this paper is excellently written with a very clear narrative that addresses all the core questions you might have with such an approach. In terms of novelty, the use of enrichment probes is not new, and the methodology chosen for that element is near identical to commercially available methods (e.g. using IDT xGen Lockdown probes). However, the clever reuse of oligo synthesis pools I have not seen before and is a highly valuable contribution allowing this approach to become feasible to labs where the same enrichment is required many times (e.g., > 100). The figures are beautifully presented and it was refreshing to see all data and code available in public repositories. I believe this work will be of great interest to the growing long-read sequencing community and an excellent fit for Nature Communications.

I had the following comments that I would like addressed:

We thank the reviewer for the highly positive comments about our work. The reviewer commented that “this work will be of great interest to the growing long-read sequencing community and an excellent fit for Nature Communications”. We have revised our manuscript to address the reviewer's questions and suggestions, as detailed below.

Line 106: Where does the >10,000 reactions come from? I understand the mixed oligo pool can be used numerous times, but I cannot see how there would be enough initial material in the non-biotinylated pool. Some elaboration on this point would be helpful and if not tested, perhaps using the term “in theory” would be more appropriate.

This number of >10,000 reactions was a ballpark estimate based on the amount of TEQUILA probes that could be generated from the input material of non-biotinylated template oligo pool. In response to the reviewer’s question and a similar question from Reviewer 1, we have performed a series of probe synthesis reactions for the 468 actionable cancer genes, using varying amounts of template oligo pool ranging from 2 ng to 10 ng, and varying incubation times ranging from 1 hr to 16 hr. For each of these combinations, TEQUILA probes were synthesized using fixed concentrations of enzymes (0.8 U/μl of Klenow Fragment (3’-5’ exo-) DNA polymerase, 3 nM (0.04 U/μl) of Nt.BspQI), 0.6 mM dNTPs, and other reaction components as described in the Methods section. We quantified the resulting probe yield using Nanodrop (Fig. R1). With increasing amounts of template oligo pool, we observed a corresponding increase in probe yield. Moreover, the probe yield increased with longer incubation times, although the rate of increase slowed down after 4 hours. This may be explained by the fact that the enzymes and dNTPs used in the probe synthesis reaction became exhausted over time.

Figure R1: Probe yield of individual synthesis reactions measured by Nanodrop. a and b show probe yield from different amounts of template oligo pool (ranging from 2 ng to 10 ng) and incubation times (ranging from 1 hr to 16 hr).

Importantly, using an input of only 2 ng of template oligo pool, we were able to generate up to 25 ug of TEQUILA probes, which can be used for 250 capture reactions, with 100 ng of probes per capture reaction. For a panel of 6,000 probes, the total amount of non-biotinylated template oligo pool purchased from commercial vendors typically ranges from 50 ng to more than 200 ng based on our experience, depending on batch variations during array-based oligo synthesis. Assuming that 2 ng of template oligo pool is regularly used per probe synthesis reaction, the total amount of template oligo pool can be used for 25 to 100 TEQUILA probe synthesis reactions. Therefore, the total amount of TEQUILA probes can potentially be used for 6,250 (250 × 25) to 25,000 (250 × 100) capture reactions. Overall, the results demonstrate a high probe synthesis yield. We have included these new results and Fig. R1 as a new supplementary figure in the revised manuscript.

Line 111: Why is PCR amplification required? Wouldn't this add more bias to the quantification step? I also find it surprising that such a good correlation is seen after the level of amplification performed. Were you not surprised? And why do you think this is?

The TEQUILA-seq protocol is based on the standard workflow of targeted RNA sequencing using biotinylated capture oligos, which includes two rounds of PCR amplification (PMID: 22081020). The first round of PCR amplification, performed before target enrichment, generates sufficient cDNA for target capture and allows us to start with a relatively low input amount of total RNA. The second round of PCR amplification, performed after target enrichment, amplifies the captured cDNA to generate sufficient cDNA for nanopore 1D cDNA library preparation and sequencing. We optimized the number of PCR cycles to avoid overamplification while meeting the required input amounts for target capture and library preparation. Specifically, we amplified cDNA by 11 cycles for the first round of PCR amplification and by 12 cycles for the second round of PCR amplification. In contrast, RNA Capture Long Seq (PMID: 29106417) amplified cDNA by 18 cycles for each round of PCR amplification, while ORF-Capture-Seq (PMID: 32393825) amplified captured cDNA by 30 cycles for the second round of PCR amplification alone.

Importantly, our results in Fig. 2 demonstrate that TEQUILA-seq preserves the abundances and proportions of transcript isoforms. To summarize, TEQUILA-seq uses a limited number of PCR cycles to avoid overamplification while generating sufficient cDNA for target capture and library preparation.

Line 133: When comparing the fold-enrichment against the whole-transcriptome sequencing, why was quantification not assessed on this data set. You later look at using the ERCC standard, but those are synthetic and not particularly long and you have the data from a more realistic setting to present? Is there a reason for this, and how well does the quantification of your enriched sample compare to the whole transcriptome dataset? (I know you later go on to look at this with other data sets, but I was confused as to why this initial one was not assessed as well?)

We did not assess transcript quantification for this small 10-gene panel, as we had thought that performing the assessment using larger gene panels (as shown in Fig. 2) would generate more reliable and informative results. Nonetheless, the reviewer's point is very well taken. For completeness, we have performed a new analysis to assess the extent to which TEQUILA-seq of the 10-gene panel preserves transcript quantification. We found that the estimated abundances of the 10 target genes by both TEQUILA-seq and xGen Lockdown-seq were highly correlated with the estimated abundances by whole-transcriptome 1D cDNA sequencing (Pearson's correlation of 0.985 and Spearman's correlation of 0.976 between TEQUILA-seq and whole-transcriptome 1D cDNA sequencing; Pearson's correlation of 0.987 and Spearman's correlation of 0.988 between xGen Lockdown-seq and whole-transcriptome 1D cDNA sequencing; see Fig. R10). We have added these results to the revised manuscript.

Figure R10: Correlation between estimated abundances of 10 brain genes by TEQUILA-seq, xGen Lockdown-seq, and whole-transcriptome 1D cDNA sequencing.

Line 320: It is fairer to say that while the setup costs are comparable to existing approaches, reagents for a specific enrichment panel can be used many times, reducing the overall costs. The costs you provide per reaction are with what I feel is a very high reuse, which in many cases (e.g., where different problems are studied) may not be true.

Yes, we agree and have toned down our language as suggested. We should also note that even with a modest number of re-use (e.g. 100), TEQUILA-seq still enables much lower per-reaction cost and higher flexibility for setup and use, as compared to commercial solutions.

To address this comment, we have updated the following text in the revised manuscript (revised text highlighted in bold below):

*“This pool can **potentially** be used to synthesize TEQUILA probes for **6,250 to 25,000 reactions**, at **\$0.31-\$0.53/reaction** when considering the costs of reagents and consumables.”*

*“By using nickase-triggered isothermal SDA, the TEQUILA process can generate large quantities of biotinylated capture oligos from limited starting material, enabling **a large number of** capture reactions **thus substantially reducing the per-reaction cost**. As the nickase releases the synthesized strand from the universal adaptor sequence, the TEQUILA probes are free of any artificial adaptor sequence, with only complementary sequences against the targeted sequences. TEQUILA reduces the initial set up cost and drastically (by 2-3 orders of magnitude) reduces the per-reaction cost of targeted capture, as compared to a standard commercial solution, **assuming a high number of re-use** (Supplementary Tables 1 and 2). With this cost structure, TEQUILA-seq can readily scale up to large cohorts with many biological samples.”*

General: A major advantage of emerging RNA sequencing approaches with long-read approaches like nanopore is the ability to directly read the modification that are present. This information is lost when using this method. It would be useful to perhaps highlight this limitation in the discussion to ensure this limitation is understood.

We appreciate the reviewer's suggestion and have added the following text to the Discussion section of the revised manuscript to highlight this limitation of TEQUILA-seq:

“Importantly, TEQUILA probes can be used for targeted sequencing in both long-read and short-read RNA-seq workflows. One limitation of TEQUILA-seq, however, is that it is designed for cDNA sequencing but is not compatible with direct RNA-seq on the ONT platform for sequencing native RNA molecules (PMID: 31740818). Therefore, TEQUILA-seq cannot take advantage of the unique features of the ONT direct RNA-seq workflow, including its capability to directly read base modifications of RNA (PMID: 36203024, 34750572, 34711425).”

Title: Given that long-read RNA sequencing is often confused with Direct RNA sequencing, it would be more appropriate for the title to be: “TEQUILA-seq: A versatile and low-cost method for targeted long-read cDNA sequencing”.

While we understand and appreciate the reviewer's perspective, we would very much like to keep the title of our paper as is. Historically, “RNA sequencing” has been a standard term for cDNA sequencing on long-read and short-read platforms. It is also arguably the much more frequently used keyword and search term than “cDNA sequencing”. By changing “RNA sequencing” to “cDNA sequencing” in the title of our paper, we are concerned that this could decrease the searchability and discoverability of our work. To the best of our knowledge, “direct RNA sequencing” has been a designated term in the literature for ONT sequencing of native RNA molecules (PMID: 34893601, 36203024). In light of the reviewer's comment, we have added new text to the Discussion section to highlight that TEQUILA-seq is not compatible with the ONT direct RNA-seq workflow (please see our response above to the reviewer's previous comment). We think that this is a more balanced approach to address the reviewer's concern and avoid potential confusions, without overcompensating for this issue in a way that could reduce the visibility of our work to a broad audience.

REVIEWERS' COMMENTS

Reviewer #1 (Remarks to the Author):

I want to thank the author for their detailed responses to all my requests. They managed to clearly address all my concerns.

I am very happy with this response. I think this will be a great addition to the field and should be published without delay.

Reviewer #2 (Remarks to the Author):

The authors have done a great job of revising this manuscript and addressing the reviewers' comments and queries. I have no further comments/concerns.

Reviewer #3 (Remarks to the Author):

I commend the authors for taking on my comments and addressing them in the revised manuscript.